# The health status alters the pituitary function and reproduction of mice in a *Cxcr2*-dependent manner

Colin Timaxian[1,2], Isabelle Raymond-Letron[3], Céline Bouclier[1], Linda Gulliver[4], Ludovic Le Corre[5], Karim Chébli[6], Anne Guillou[7], Patrice Mollard[7], Karl Balabanian[2,8], Gwendal Lazennec[1,2]

**Microbiota and chronic infections can affect not only immune status, but also the overall physiology of animals. Here, we report that chronic infections dramatically modify the phenotype of *Cxcr2* KO mice, impairing in particular, their reproduction ability. We show that exposure of *Cxcr2* KO females to multiple types of chronic infections prevents their ability to cycle, reduces the development of the mammary gland and alters the morphology of the uterus due to an impairment of ovary function. Mammary gland and ovary transplantation demonstrated that the hormonal contexture was playing a crucial role in this phenomenon. This was further evidenced by alterations to circulating levels of sex steroid and pituitary hormones. By analyzing at the molecular level the mechanisms of pituitary dysfunction, we showed that in the absence of Cxcr2, bystander infections affect leukocyte migration, adhesion, and function, as well as ion transport, synaptic function behavior, and reproduction pathways. Taken together, these data reveal that a chemokine receptor plays a direct role in pituitary function and reproduction in the context of chronic infections.**

## Introduction

Bystander chronic infections are common in rodent animal conventional facilities with a high prevalence of viruses such as mouse norovirus, parvovirus, mouse hepatitis virus, rotavirus, and bacteria such as helicobacter (Pritchett-Corning et al, 2009). Because of the possible deleterious effects of such infections, this has led to a recent trend of rethinking of the health status of animal facilities and the development of Specific and Opportunistic Pathogen-Free (SOPF) or of specific pathogen-free (SPF) animal facilities to limit the influence of the environment on the phenotype of mice, especially in the case of immune or inflammatory studies. In SOPF

conditions, mice are devoid of both pathogens and opportunistic infections, whereas in SPF conditions, they are devoid of pathogens only. However, there is quite a debate about using pathogen-free mice as animal models because several reports have shown that mice exposed to bystander infections better recapitulate the human immune situation than mice housed in pathogen-free conditions (Beura et al, 2016) and that infections can affect the response to vaccination (Reese et al, 2016). Moreover, it has also been shown that transplanting C57BL/6 embryos into domestic wild-type mice trapped in horse stables better recapitulate human immune response than laboratory animals, reinforcing the importance of microbiota (Rosshart et al, 2019). Such pathogen-free influences could also account for some of the difficulties in translating animal studies into treatments for patients. It remains that genetic alterations produced in mouse models frequently do not lead to the same phenotypes as those observed in humans with similar alterations. One relevant example is the response to infection (Cypowyj et al, 2012). Infections are transmitted through different generations of animals, in particular during birth, but also during co-housing and breast feeding (McCafferty et al, 2013).

The role of microbiota is not only important in the context of immune studies (Hooper et al, 2012; Honda & Littman, 2016) but also can affect the outcome of different pathologies such as inflammatory bowel disease (Bloom et al, 2011), Crohn's disease (Cadwell et al, 2010), atherosclerosis (Wang et al, 2015), arthritis (Scher et al, 2013), asthma (Thorburn et al, 2015), or cancer (Roy & Trinchieri, 2017). Importantly, the genotype of mouse models does not contribute to the totality of phenotype observed and can be largely influenced by the various types of microbiota, within some cases, a greater impact of the microbiota than the genotype on the phenotype. This has led to the concept of "host gene plus microbe" or metagenome (Stappenbeck & Virgin, 2016). For these reasons, the use of SPF or SOPF husbandry can be viewed as an excellent way to normalize experiments and to limit the inter-individual or inter-housing variability and to improve the reproducibility of the results. However, factors other than microbiota can also deeply affect the

[1]Centre National de la Recherche Scientifique (CNRS), SYS2DIAG-ALCEDIAG, Cap Delta, Montpellier, France   [2]CNRS, Groupement de Recherche 3697 "Microenvironment of Tumor Niches," Micronit, France   [3]Department of Histopathology, National Veterinary School of Toulouse, France and Platform of Experimental and Compared Histopathology, STROMALab, Unité de recherche mixte (UMR) Université Paul Sabatier/CNRS 5223, Etablissement français du sang, Institut national de la santé et de la recherche médicale (Inserm) U1031, Toulouse, France   [4]University of Otago, Dunedin, New Zealand   [5]Nutrition et Toxicologie Alimentaire (NUTOX) Laboratory - INSERM Lipides, Nutrition, Cancer UMR 1231 - AgrosupDijon, Dijon, France   [6]Equipe Metazoan Messenger RNAs Metabolism, Montpellier, France   [7]Institut de Génomique Fonctionnelle, CNRS, INSERM, University of Montpellier, Montpellier, France   [8]Université de Paris, Institut de Recherche Saint-Louis, EMiLy, INSERM U1160, Paris, France

Correspondence: gwendal.lazennec@sys2diag.cnrs.fr

phenotype of mouse models, including husbandry conditions, such as temperature, light–dark cycles, diet, water, noise, hygrometry, and handling of animals by care takers. Nevertheless, the microbiota is sometimes necessary to generate the phenotype. Indeed, a mouse model of Crohn's disease with mice harboring a mutation in Atg16/1 gene, showed the expected phenotype in conventional conditions but not in SPF housing (Cadwell et al, 2008, 2010). On the other hand, bystander infections can lead to a loss of a particular phenotype, such as in some models of diabetes (Bach, 2002; Okada et al, 2010). This can be complicated further by the fact that the nature of microbiota can lead to different phenotypes, as exemplified in another model of Crohn's disease with mice deficient for Nod2 (Ramanan et al, 2014, 2016).

Because one of the primary effects of bystander infections will be alterations to the immune system and the inflammation process (Tao & Reese, 2017), particular attention should be paid to pro-inflammatory cytokines and in particular chemokines. Chemokines are chemotactic cytokines of 60–100 amino acids that can be divided into four subtypes (CXC, CC, C, or CX3C), based on the location of cysteines in the N terminus of the protein (Zlotnik & Yoshie, 2000). Chemokines are ligands of seven transmembrane G$\alpha$i protein-coupled receptors, signaling in particular through the phosphatidylinositol-3 kinase (PI3K)/Akt, PLC/PKC and MAPK/p38, Ras/Erk and JAK2/signal transducer, and activator of transcription (STAT3) pathways (Wang & Knaut, 2014). Chemokines and their receptors play a major role in the trafficking of immune cells, notably during immune reaction or inflammatory events (McCully et al, 2018), but their role is not restricted to immune processes, as they have been reported to be important in a number of other physiologic or pathologic events. These include angiogenesis (Strieter et al, 2005b), metabolism (Chavey et al, 2009), chronic obstructive pulmonary disease (Henrot et al, 2019), neurodegenerative disease, and cancer (Lazennec & Richmond, 2010; Lazennec & Lam, 2016). Among chemokine receptors, Cxcr2, which is expressed in neutrophils and endothelial cells, appears essential in the control of angiogenesis, through the binding of E (glutamate), L (leucine), R (arginine) (ELR)-motif containing chemokines (CXCL1, CXCL2, CXCL3, CXCL5, CXCL6, CXCL7, and CXCL8). ELR-motif chemokines harbor the tripeptide glutamic acid–leucine–arginine motif present in the N-terminal part of the protein (Strieter et al, 2005a). Cxcr2 regulates wound healing (Devalaraja et al, 2000), angiogenesis (Addison et al, 2000), multiple sclerosis (Liu et al, 2010), Alzheimer's disease (Tsai et al, 2002), atherosclerosis (Boisvert et al, 2000), respiratory diseases (Strieter et al, 2005b), resistance to infections (Cummings et al, 1999), and is involved in cancer (Freund et al, 2003; Ali & Lazennec, 2007; Bieche et al, 2007; Lazennec & Richmond, 2010). Cxcr2 KO animals exhibit splenomegaly due to an increased number of metamyelocytes and neutrophils, and impairment in the recruitment of neutrophils during acute inflammatory conditions (Cacalano et al, 1994).

Here, we report that the action of microbiota on mouse phenotype is dependent on the absence of Cxcr2 protein. In the absence of Cxcr2, mice are clearly affected by the presence of pathogens. However, in the absence of pathogens, Cxcr2 KO mice display a similar external phenotype to that of wild-type (WT) mice in terms of their ability to reproduce and their gross appearance (Cacalano et al, 1994; Broxmeyer et al, 1996). By contrast, in conditions of bystander infections, Cxcr2 null mice exhibit an impaired

reproductive ability and reduced development of reproductive organs. Using mammary gland and ovary transplant experiments, we show that reproductive function can be restored to Cxcr2 KO mice in a WT context, despite the presence of pathogens. We also show that the absence of Cxcr2 not only leads to susceptibility to infection but also leads to reproductive defects due to major impairment of pituitary function controlling the production of pituitary hormones. This study therefore reveals a novel role for the chemokine receptor Cxcr2 in pituitary physiology, which has been discovered in the context of microbiota infections. This has never been reported for any chemokine receptor.

# Results

We have been working for a long time on Cxcr2 ligands (Freund et al, 2003, 2004; Bieche et al, 2007) and we wished to use Cxcr2 KO animals to analyze its role in vivo. Our study started with the serendipitous finding that Cxcr2 KO animals had distinct breeding abilities in conventional or SOPF animal facilities. To evaluate the possible action of microbiota mouse phenotype in the context of Cxcr2 deficiency, mice were housed either in an SOPF animal facility in sterile conditions or in a conventional animal facility with possible bystander infections. In SOPF conditions, Cxcr2 KO animals displayed the same breeding ability as WT animals, confirming prior work of Cacalano et al (1994) (Fig 1A).

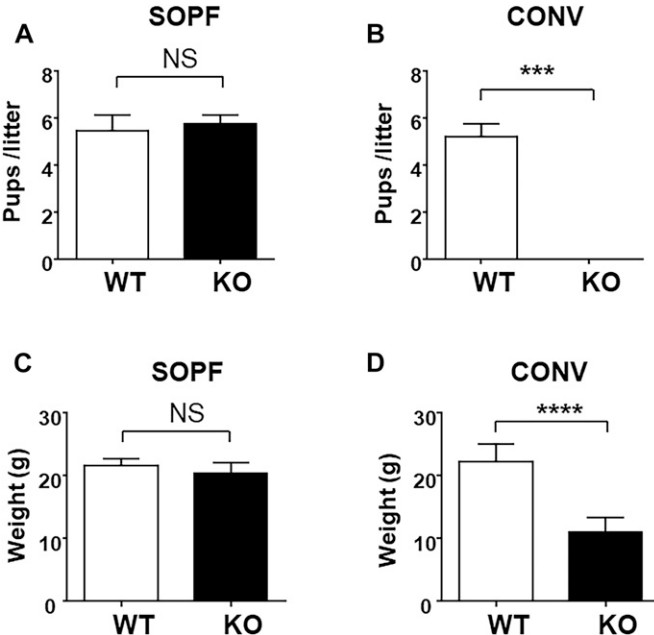

**Figure 1. Husbandry in conventional housing conditions alters the reproduction of *Cxcr2* KO animals.**
**(A)** Number of animals/litter issued in the breeding of WT animals or *Cxcr2* KO animals in SOPF conditions. The data represent the mean ± SEM of at least 10 matings (Mann–Whitney test, NS, nonsignificant, *$P < 0.05$, **$P < 0.01$, ***$P < 0.001$). **(B)** Same breeding experiment in conventional conditions with bystander infections. The results represent the mean ± SEM of at least eight animals. **(C)** Weight of 12-wk-old female mice in SOPF conditions. **(D)** Weight of 12-wk-old female mice in conventional conditions. Data represent the mean ± SEM of at least eight animals. (Mann–Whitney test, ****$P < 0.0001$).

On the contrary, after the transfer of SOPF animals in conventional conditions, we observed that after several generations in conventional conditions, *Cxcr2* KO animals lost their breeding ability. Both males and females were infertile (Fig 1B), even when mated with WT animals. Of particular note, the products of such mating, *Cxcr2* heterozygous animals, were much less affected by these conditions and could breed nearly normally (although with a delayed time for successful breeding) and generate *Cxcr2* KO animals (data not shown). Moreover, *Cxcr2* KO females of conventional conditions exhibited a smaller weight than WT animals (Fig 1D), which was not the case in SOPF conditions (Fig 1C). Screening of bystander infections of animals housed in conventional conditions showed the presence of mouse norovirus, helicobacter, and *Entamoeba* sp. (Fig S1). We hypothesized that these infections were responsible for the observed phenotype as rederivating conventional animals to remove pathogens led to animals with full reproduction ability. Moreover, this combination of pathogens was not specific for the phenotype observed, as housing *Cxcr2* KO animals in other conventional facilities with another set of pathogens (including mouse norovirus, mouse hepatitis virus, other strains of helicobacter, or pinworms) led to the same results (Fig S1).

To understand why *Cxcr2* KO animals were infertile in conventional conditions, we decided to focus on females. We first performed vaginal smears of WT and KO animals of conventional conditions (Fig 2A). This showed that WT animals displayed a classical cycling with proestrus, estrus, metestrus, and diestrus. On the other hand, KO animals displayed mixed populations of cells, with no real homology to any steps of estrus cycling, suggesting that the mice were not cycling. In SOPF conditions, WT and KO animals displayed a normal estrus cycle (Fig S2). To assess the functionality of the ovary, we analyzed the ovaries of WT and KO animals from conventional or SOPF conditions. We observed that in SOPF conditions, both WT and KO animals displayed a normal histology of the ovary, with all stages of follicle maturation and the presence of multiple corpora lutea, suggesting that the mice were able to ovulate (Fig 2B, upper panel). On the contrary, in conventional conditions, whereas the ovary of the WT animals appeared to have completely normal histologic appearance, ovaries from KO animals displayed a large number of follicles in all stages of development including atresia, but did not exhibit any corpus luteum, suggesting that KO *Cxcr2* mice could not ovulate (Fig 2B, lower panel).

We next looked at other reproductive organs, including uterus and mammary gland. The uterus of WT and KO animals in SOPF conditions appeared normal and of similar size (Fig 3A, upper panel). In conventional conditions, the uterus of KO animals was smaller in diameter compared with WT animals and was in a rest status, whereas WT uterus was cycling (Fig 3A lower panel). Maximum uterine thickness and the external uterine diameter were approximately fourfold reduced in KO animals compared with WT animals (Fig S3A). Whereas uteri of WT mice had well-defined layers, uterine layers were less discernible in KO animals, appearing compressed and very cellular. KO mice also showed a loss of the normal convoluted appearance of the uterine luminal epithelium, assuming a more linear profile. The endometrium thickness, the luminal epithelium thickness, and the number of glandular lumen were also decreased in KO animals, suggesting a noncycling uterus (Fig S3B).

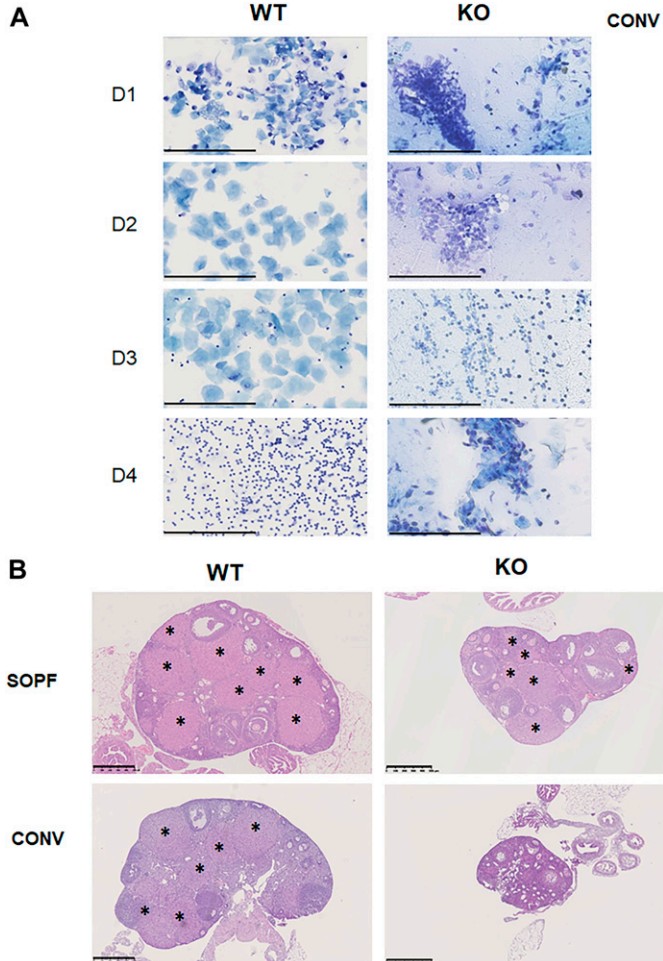

**Figure 2.  *Cxcr2* KO animals in conventional housing conditions exhibit cycle defects and altered ovary morphology.**
**(A)** Representative bright-field microscopic images (40×) for Giemsa-stained vaginal smears from different estrous cycle stages of WT and KO animals housed in conventional conditions. For WT animals, day 1: proestrus (mostly nucleated epithelial cells), day 2: estrus (cornified epithelial cells), day 3: metestrus (cornified epithelial cells with leukocytes), and day 4: diestrus (mostly leukocytes). For KO animals, the content of vaginal smears, no clear state of estrus cycle could be determined. Scale bars: 200 *μm*. **(B)** Histology of the ovary of WT and KO animals in SOPF (upper part) or conventional conditions (lower part). Representative images of hematoxylin-eosin stained ovaries at a 5× magnification are shown here. Stars indicate the presence of corpus lutea. Scale bars: 500 *μm*.

We also observed an altered morphology of the mammary gland in KO animals in conventional conditions. Whole-mount experiments showed a complete branching in WT animals, whereas the mammary gland of KO animals displayed a rudimentary branching (Fig 3B). Mammary glands of KO appeared to have a significant reduction in the numbers of glandular (ductal) profiles (Fig S3C, lower panel). Epithelial cells lining ducts in KO mice often appeared haphazardly arranged and the mammary gland than its WT counterpart. The situation appeared different in the mammary gland of SOPF animals with a similar branching in WT and KO animals (Fig S3C, upper panel).

To understand the reasons for the reproductive defects in KO animals housed in conventional conditions, we decided to first compare the transcriptomic profiles of the mammary gland of WT

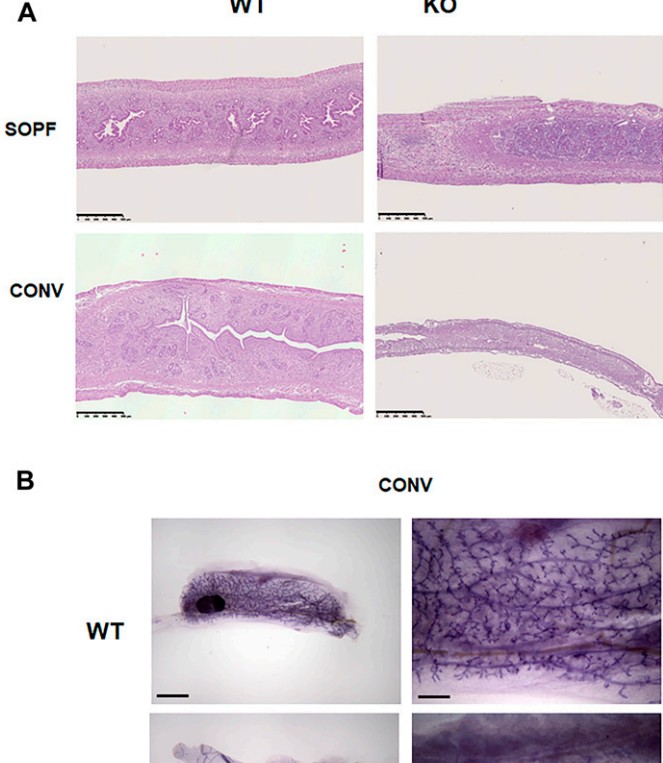

**Figure 3.   The uterus and mammary gland of Cxcr2 KO animals show defects in conventional conditions.**
**(A)** Histology of the uterus of WT and KO animals in SOPF (upper part) or conventional conditions (lower part). Representative images of hematoxylin–eosin–stained uteri at a 5× magnification are shown here. Scale bars: 500 μm. **(B)** Whole mount of mammary glands from 13 wk WT and KO animals in conventional conditions. Scale bars: 5 mm (left panel) or 1.3 mm (right panel).

and KO animals in SOPF with those in conventional conditions by RNAseq. Principal component analysis of RNAseq showed that mammary glands of WT and KO animals were close to each other, whereas the one of WT and KO conventional animals were more widely distributed (Fig S4). We observed that the transcriptome of the mammary gland of KO animals was much more altered in conventional than that of SOPF conditions, with ~10-fold more genes up-regulated in WT animals versus KO animals (Fig 4A). Among the genes up-regulated in the mammary gland of WT compared with KO animals, only 12 were common between conventional and SOPF conditions (Fig 4B), and only four were common for down-regulated genes. Gene ontology (GO) analysis showed that the common down-regulated genes were essentially related to leukocyte chemotaxis and migration, and host defense (Fig 4C). In SOPF conditions, the genes down-regulated in the mammary gland of KO animals were essentially those linked to muscle development and differentiation (Fig 4D and Table 1), whereas the genes up-regulated in KO mice were related to granulocyte migration and leukocyte aggregation/adhesion (Fig 4E and Table 2). We next focused

on the major alterations of the transcriptome of the mammary gland of KO animals in conventional conditions. GO analysis showed that most of the genes down-regulated in the mammary gland of KO animals were related to three major biological processes (Table 3): mammary gland development and differentiation (Fig 5A), epithelial cell proliferation (Fig 5B), and Wnt signaling (Fig 5C). On the other hand, genes up-regulated in the mammary gland of KO animals in conventional conditions (Table 4) were involved in leukocyte migration (Fig 5D) and muscle function (Fig 5E). The RNAseq data were validated by real-time PCR on a subset of representative genes of the different GO identified or known to play a role in mammary gland physiology. We observed a common up-regulation of S100a8, S100a9, Mmp8, and Ngp genes related to chemotaxis in the mammary gland of KO animals in conventional and SOPF conditions (Fig 6). On the other hand, Areg, lactoferrin, CXCL15, Elf5, Sox10, Ido1, Wnt2, Prlr, Krt15, and Gata3 were down-regulated only in the mammary gland of KO animals in conventional conditions (Fig 6). Of particular note, Areg, Lactoferrin, Wnt2, Prlr, Elf5, and Gata3 are genes known to be critical for development of the mammary gland according to the studies performed with KO animals for these genes (Luetteke et al, 1999; Kelly et al, 2002; Zhou et al, 2005; Kouros-Mehr et al, 2006; Watson & Khaled, 2008).

We also analyzed the differences in the ovary of WT and KO animals in conventional conditions, by looking at some key genes known to play a role in ovary function. We report a decrease in the expression of Akrc18, Cyp19, Hsd3b2, Prlr, and lactoferrin genes in the ovary of KO animals, whereas AR expression was strongly induced (Fig 7A). Akr1c18 encodes 20α-hydroxysteroid dehydrogenase, a progesterone-metabolizing enzyme (Piekorz et al, 2005). Cyp19 or estrogen synthase is an aromatase of the P450 family involved in, in particular, the aromatization of androgens to estrogens (Rosenfeld et al, 2001). Hsd3b2 encodes hydroxy-delta-5-steroid dehydrogenase, 3 beta-, and steroid delta-isomerase 2, which is involved the conversion of 5-ene-3β-hydroxysteroids to 4-ene-3-ketosteroid, an essential step in the biosynthesis of progesterone and estrogens in the ovary (Payne et al, 1995). Interestingly, many of these enzymes are regulated by prolactin (PLR), and prolactin receptor (Prlr) is critical (Bachelot & Binart, 2005; Stocco et al, 2007). Androgen receptor (Ar) also plays a critical role in ovary function, and Ar KO leads to premature ovarian failure (Shiina et al, 2006; Walters, 2015). The alteration of these key regulatory genes in the ovary suggested to us a possible impairment of hormone production. We thus measured progesterone and estradiol serum levels in WT and KO animals in conventional conditions. In agreement with the absence of corpus luteum in KO ovaries, we observed a decrease in progesterone levels relative to WT (Fig 7B). On the other hand, estradiol levels were increased.

This led us to hypothesize that an alteration of the hormonal context could explain the reproductive defects observed in KO animals housed under conventional conditions. To test this, we performed ovary transplantation experiments (Fig 7C). WT mice were ovariectomized and a WT or KO ovary was reimplanted within the oviduct bursa (Behringer, 2017). We observed that both WT and KO transplanted ovaries were able to display a normal phenotype with the presence of corpora lutea (Fig 7C). Moreover, when transplanted females were bred, they were able to give birth (Fig 7C). To confirm the role of the hormonal environment in the KO defects, we also performed mammary gland transplantation (Fig 7D). The

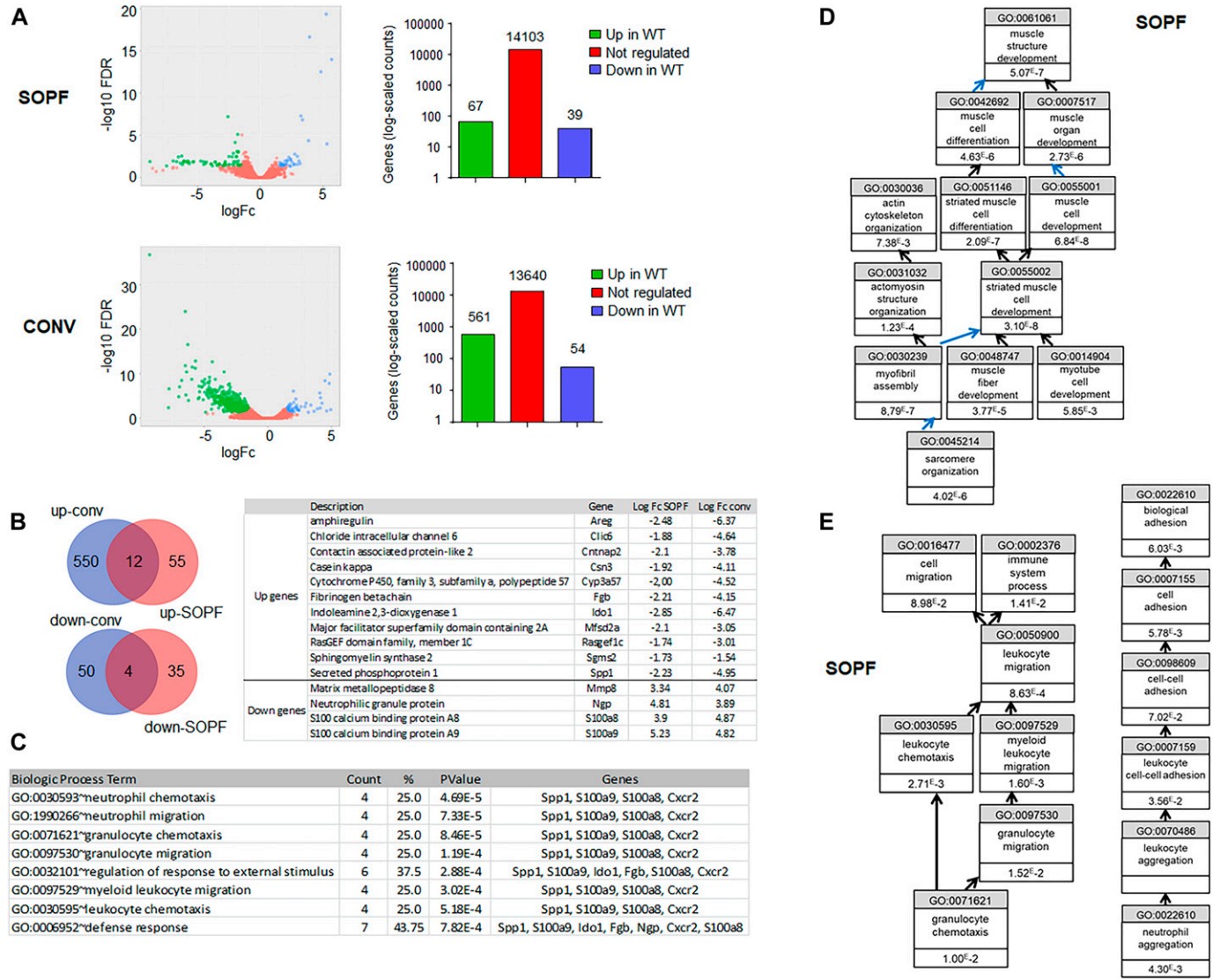

**Figure 4. Differential gene expression in WT and KO mammary glands is more pronounced in conventional conditions.**
**(A)** Left panel: the volcano plots show the global changes in RNA expression patterns for WT versus KO mammary glands in SOPF or conventional conditions. Data represent analysis of cpm estimates with a log of fold change of more than 1.5-fold change and $P < 0.05$ of 3 animals per group. Right panel: number of differentially regulated genes for the same analysis. **(B)** Left panel: Venn diagram representing the common genes up-regulated or down-regulated in the mammary gland of WT compared with KO animals in conventional versus SOPF conditions. Right panel: list of common genes. **(C)** Gene Ontology (GO) analysis of biological process for the common genes regulated in KO animals in conventional and SOPF conditions. **(D)** GO analysis of biological process of down-regulated genes in KO mammary glands of SOPF is mostly related to muscle function. Black arrows mean "is a." Blue arrows mean is "part of." **(E)** GO analysis of biological process of up-regulated genes in KO mammary glands of SOPF are mostly related to granulocyte chemotaxis (left panel) and to neutrophil aggregation (right panel).

mammary fat pads of young WT female mice were cleared of all epithelial structures and either reimplanted with WT or KO mammary gland fragments or left untreated. Control mammary glands without transplant did not develop any ductal branching, whereas both WT and KO transplants could fully restore a functional mammary gland (Fig 7D). Together, these data suggest that the hormonal environment of WT animals is sufficient to enable the KO ovary and mammary gland to be functional. As steroid hormone production is controlled by the pituitary, we assessed the serum levels of the pituitary hormones follicle stimulating hormone (FSH), luteinizing hormone (LH), PRL, and growth hormone (GH) in WT and KO animals in conventional conditions. We report that the four

pituitary hormones tested displayed a clear decrease in KO animals (Fig 8A), suggesting major defects in the pituitary function of KO animals housed under conventional conditions. We did not measure the hormone levels of transplanted animals (Fig 7C), as these mice were used for breeding and could not be compared with virgin animals.

To understand at the molecular level, the reasons for the pituitary dysfunction in KO animals, we performed an RNAseq analysis of pituitary glands from WT and KO animals in SOPF and conventional conditions (Fig 8B). Strikingly, very little difference was observed between the pituitaries of WT and KO animals in SOPF conditions. On the other hand, more than 850 genes were either up-regulated or down-regulated in the pituitary of KO animals in

**Table 1.   Most enriched pathways for genes down-regulated in the mammary gland of KO versus WT specific and opportunistic pathogen-free animals.**

| GO biological process term | Count | % | *P*-value | Genes |
|---|---|---|---|---|
| GO:0003012~muscle system process | 15 | 23.43 | 3.92E-12 | Cmya5, Mybpc2, Tmod1, Ttn, Tcap, Srl, Atp2a1, Myl1, Hrc, Ryr1, Casq1, Actn3, Tnni2, Cacna1s, and Myom1 |
| GO:0006936~muscle contraction | 13 | 20.31 | 3.40E-11 | Mybpc2, Tmod1, Ttn, Tcap, Atp2a1, Myl1, Hrc, Ryr1, Casq1, Actn3, Tnni2, Cacna1s, and Myom1 |
| GO:0006941~striated muscle contraction | 9 | 14.06 | 1.60E-8 | Tnni2, Ttn, Tcap, Atp2a1, Myl1, Hrc, Casq1, Actn3, and Cacna1s |
| GO:0055002~striated muscle cell development | 9 | 14.06 | 3.12E-8 | Actn3, Tmod1, Ttn, Cacna1s, Tcap, Ldb3, Ryr1, Casq1, and Neb |
| GO:0055001~muscle cell development | 9 | 14.06 | 6.84E-8 | Actn3, Tmod1, Ttn, Cacna1s, Tcap, Ldb3, Ryr1, Casq1, and Neb |
| GO:0051146~striated muscle cell differentiation | 10 | 15.62 | 2.09E-7 | Actn3, Tmod1, Ttn, Cacna1s, Smyd1, Tcap, Ldb3, Ryr1, Casq1, and Neb |
| GO:0090257~regulation of muscle system process | 9 | 14.06 | 2.37E-7 | Actn3, Tnni2, Cmya5, Ttn, Srl, Atp2a1, Ryr1, Hrc, and Casq1 |
| GO:0061061~muscle structure development | 13 | 20.31 | 5.07E-7 | Jph1, Tmod1, Ttn, Smyd1, Tcap, Neb, Casq1, Ryr1, Actn3, Cacna1s, Mylpf, Jph2, and Ldb3 |
| GO:0030239~myofibril assembly | 6 | 9.37 | 8.79E-7 | Tmod1, Ttn, Tcap, Ldb3, Casq1, and Neb |
| GO:0007517~muscle organ development | 10 | 15.62 | 2.73E-6 | Actn3, Jph1, Ttn, Cacna1s, Jph2, Mylpf, Smyd1, Tcap, Ryr1, and Casq1 |
| GO:0044057~regulation of system process | 11 | 17.18 | 2.91E-6 | Actn3, Tnni2, Cmya5, Ttn, Fgb, Cck, Srl, Atp2a1, Ryr1, Hrc, and Casq1 |
| GO:0045214~sarcomere organization | 5 | 7.81 | 4.02E-6 | Ttn, Tcap, Ldb3, Casq1, and Neb |
| GO:0042692~muscle cell differentiation | 10 | 15.62 | 4.63E-6 | Actn3, Tmod1, Ttn, Cacna1s, Smyd1, Tcap, Ldb3, Ryr1, Casq1, and Neb |
| GO:0003009~skeletal muscle contraction | 5 | 7.81 | 5.63E-6 | Actn3, Tnni2, Tcap, Atp2a1, and Casq1 |

conventional conditions (Fig 8B). Principal component analysis of RNAseq showed that pituitary of WT and KO animals were close to each other, whereas the one of WT and KO conventional animals were very different (Fig S4). According to GO analysis, the up-regulated pathways in the KO pituitaries were related to immune cell activation (in particular lymphocyte), leukocyte adhesion, and neutrophil motility and extravasation (Fig 8C and Table 5). On the other hand, down-regulated genes involving biological processes included those involved in ion transport and synaptic function (Fig 8D and Table 6), as well as control of ovarian function (Fig 8E and Table 6). To validate these data, we analyzed the expression of a set of genes representative of the different GO mentioned above by real-time PCR on a larger number of animals (Fig 9). *Elane, S100a8, S100a9, Mpo, Ngp, MMP8, Ltf,* and Serpina3an were strongly up-regulated in the pituitary of KO animals in conventional conditions and modestly or not regulated at all, in the pituitary of KO animals in SOPF conditions. Elane, S100a8, and S100a9 are involved in migration, adhesion, and immune response. *Mpo, Ngp, Ltf,* and Serpina3an are contributing to migration and immune response. In contrast, *Prl, Crhbp, Akr1c14, Vip,* and *Rln* were all down-regulated in the pituitary of conventionally housed KO animals but not in SOPF conditions (Fig 9). *Prl, Crhbp,* and *Vip* are involved in ion transport, behavior and reproduction. *Reln* and *Crhbp* play a role in ion transport, synapse function and behavior.

## Discussion

In this study, we investigated the effects of bystander infections on the physiologic role of the chemokine receptor Cxcr2 using *Cxcr2-*

null mice. Our results have demonstrated that when exposed to common infections found in animal facilities, *Cxcr2* KO mice, but not WT mice, exhibit reproductive defects. Of particular note, Cxcr2 ligand levels were not altered in the mammary gland and the pituitary gland of WT animals in conventional conditions compared with SOPF conditions, according to RNAseq analysis. Homozygous males and females were both sub-fertile and females displayed alterations to their secondary sex organs. The first observation accounting for this sub-fertility was that female *Cxcr2* KO mice housed in conventional conditions were not able to cycle normally. None of the characteristic phases of the estrous cycle could be identified in these animals by vaginal smear, with mixed populations of vaginal cells suggesting a defect in ovarian function. Microscopic observation of ovaries from *Cxcr2* KO animals housed in conventional conditions showed reductions of the size of the ovary and an absence of corpora lutea, whereas in SOPF conditions, both WT and *Cxcr2* KO ovaries showed normal histology with all stages of follicle development and corpora lutea present. Interestingly, a number of genes involved in ovarian function and production of steroid hormones (*Akr1c18, Cyp19, Hsd3b2, Prlr, lactoferrin,* and *Ar*) were down-regulated in the ovary of *Cxcr2* KO animals in conventional conditions. *Akr1c18* (20-alpha-hydroxysteroid dehydrogenase), which catabolizes progesterone into 20-alpha-dihydroprogesterone (inactive steroid) is necessary for the maintenance of pregnancy (Choi et al, 2008) and KO of this gene leads to longer duration of estrous cycle and a reduced number of pups (Ishida et al, 2007). Knocking down *Cyp19* (Aromatase P450) has been shown to lead to mice lacking corpus luteum in ovary, accompanied by total infertility (Toda et al, 2001). *Hsd3b2* (hydroxy-delta-5-steroid

**Table 2. Most enriched pathways for genes up-regulated in the mammary gland of KO versus WT specific and opportunistic pathogen-free animals.**

| GO biological process term | Count | % | P-value | Genes |
|---|---|---|---|---|
| GO:0002523~leukocyte migration involved in inflammatory response | 3 | 7.5 | 2.66E-4 | S100a8, S100a9, and Elane |
| GO:0050900~leukocyte migration | 5 | 12.5 | 8.63E-4 | S100a8, S100a9, Elane, Thbs1, and Calca |
| GO:0052547~regulation of peptidase activity | 5 | 12.5 | 1.53E-3 | S100a8, S100a9, Thbs1, Wfdc18, and Ngp |
| GO:0097529~myeloid leukocyte migration | 4 | 10.0 | 1.60E-3 | S100a8, S100a9, Thbs1, and Calca |
| GO:0030595~leukocyte chemotaxis | 4 | 10.0 | 2.71E-3 | S100a8, S100a9, Thbs1, and Calca |
| GO:0044707~single-multicellular organism process | 17 | 42.5 | 3.79E-3 | Col9a3, Mmp8, Mpo, Igf2, Krt10, Elane, Muc4, Slc5a1, Gjb2, Thbs1, Calca, Irx4, S100a9, Mfap4, Rbp1, Ngp, and S100a8 |
| GO:0006952~defense response | 8 | 20.0 | 3.99E-3 | S100a9, Mpo, Igf2, Elane, Thbs1, Ngp, S100a8, and Calca |
| GO:0070488~neutrophil aggregation | 2 | 5.0 | 4.30E-3 | S100a8 and S100a9 |
| GO:0007155~cell adhesion | 8 | 20.0 | 5.78E-3 | S100a9, Igf2, Elane, Thbs1, S100a8, Calca, Mfap4, and Muc4 |
| GO:0022610~biological adhesion | 8 | 20.0 | 6.02E-3 | S100a9, Igf2, Elane, Thbs1, S100a8, Calca, Mfap4, and Muc4 |
| GO:0060326~cell chemotaxis | 4 | 10.0 | 6.04E-3 | S100a8, S100a9, Thbs1, and Calca |

dehydrogenase, 3 beta-, and steroid delta-isomerase 2) plays a crucial role in the biosynthesis of many steroids (Chapman et al, 2005). *Prlr* (prolactin receptor) KO animals are completely infertile with irregular estrous cycles (Horseman et al, 1997). *Lactotransferrin*, present in the reproductive tracts of rodents, can regulate secretory function and plays a role in fertilization (Yanaihara et al, 2007). *Ar* (androgen receptor) KO mice show a marked reduction in follicular maturation at maturity, with fewer corpora lutea in their ovaries (Hu et al, 2004). Not surprisingly then, morphometry, using multiple histological sections in the present study, identified alterations to

**Table 3. Most enriched pathways for genes down-regulated in the mammary gland of KO versus WT conventional animals.**

| GO biological process term | Count | % | P-value | Genes |
|---|---|---|---|---|
| GO:0007155~cell adhesion | 37 | 6.70 | 1.61E-8 | Ptprf, Fat2, Dscam, Perp, Fbln7, Cntnap2, Atp1b1, Cd24a, Ptk7, Epha1, Fn1, Pkp1, Lamc2, Col7a1, Lama1, Col13a1, Grhl2, Nrxn3, Itgb6, Fermt1, Spp1, Tenm2, Itgb4, Cd9, Celsr2, Cadm4, Cdh3, Cdh11, Col16a1, Col8a1, Cdh1, Itga8, Ephb1, Spon1, Nectin4, Flrt2, and Col14a1 |
| GO:0042060~wound healing | 16 | 2.89 | 1.84E-8 | Dsp, Arhgef19, Timp1, Tgfa, Msx2, Bnc1, Cdh3, Wnt5b, Tgfb3, Pak1, Plau, Ptk7, Erbb2, Fn1, Epb41 l4b, and Tgfb2 |
| GO:0008285~negative regulation of cell proliferation | 32 | 5.79 | 2.51E-8 | Ptprf, Timp2, Tfap2b, Irf6, Gata3, Sfrp1, Hspa1a, Sox9, Tfap2a, Runx1, Bnipl, Bmp7, Tgfb2, Vdr, Sfrp4, Scin, Cd9, Lif, Fgfr2, Msx2, Wnk2, Sfrp2, Frzb, Slit2, Tgfb3, Ptprz1, Plk5, Ovol2, Ror2, Nos1, Rerg, and Sox4 |
| GO:0090090~negative regulation of canonical Wnt signaling pathway | 14 | 2.53 | 2.33E-6 | Sox10, Nkd2, Cthrc1, Sfrp2, Wnt5b, Dkk3, Sfrp1, Frzb, Lrp4, Cdh1, Sox9, Ror2, Wnt4, and Sfrp4 |
| GO:0007275~multicellular organism development | 53 | 9.60 | 3.22E-6 | Shroom3, Sfrp1, Ngef, Lrp4, Enah, Tbx3, Dbn1, Sfrp4, Prrx2, Ephb3, Plekhb1, Lmx1b, Celsr2, Dkk3, Frzb, Slit2, Wnt2, Ovol2, Anpep, Wnt4, Ano1, Tmem100, Grem2, Elf3, Foxa1, Dact2, Fzd7, Wnt5b, Irx4, Kdf1, Fzd10, Mdfi, Bmp7, Mycbpap, Sema3d, Col13a1, Wnt7b, Vdr, Cited1, Smpd3, Msx2, Sfrp2, Itga8, Krt8, Eya2, Irx3, Trp63, Alx4, Ror2, Cxcl17, Dmbt1, Flrt2, and Islr2 |
| GO:0061180~mammary gland epithelium development | 6 | 1.08 | 4.66E-6 | Wnt2, Atp2c2, Prlr, Msx2, Wnt4, and Wnt7b |
| GO:0008284~positive regulation of cell proliferation | 34 | 6.15 | 5.53E-6 | Tfap2b, Ccnd1, Ptn, Cxcr2, Sfrp1, Sox9, Tbx3, Pgr, Plau, Epcam, Epha1, Erbb2, Fn1, Areg, Lamc2, Tgfb2, Akr1c18, Wnt7b, Gas1, Lif, Tgfa, Timp1, Rab25, Fgfr2, Sfrp2, Wnt2, Id4, Pak1, Efemp1, Folr2, Osr2, Cldn7, Klf5, Sox4 |
| GO:0030855~epithelial cell differentiation | 11 | 1.99 | 8.26E-6 | Krt14, Muc1, Upk2, Trp63, Elf3, Aldoc, Vil1, Fgfr2, Bmp7, Bdh2, and Ehf |
| GO:0045669~positive regulation of osteoblast differentiation | 11 | 1.99 | 9.46E-6 | Id4, Cd276, Trp63, Cthrc1, Msx2, Bmp7, Sfrp2, Wnt4, Ltf, Wnt7b, and Fbn2 |

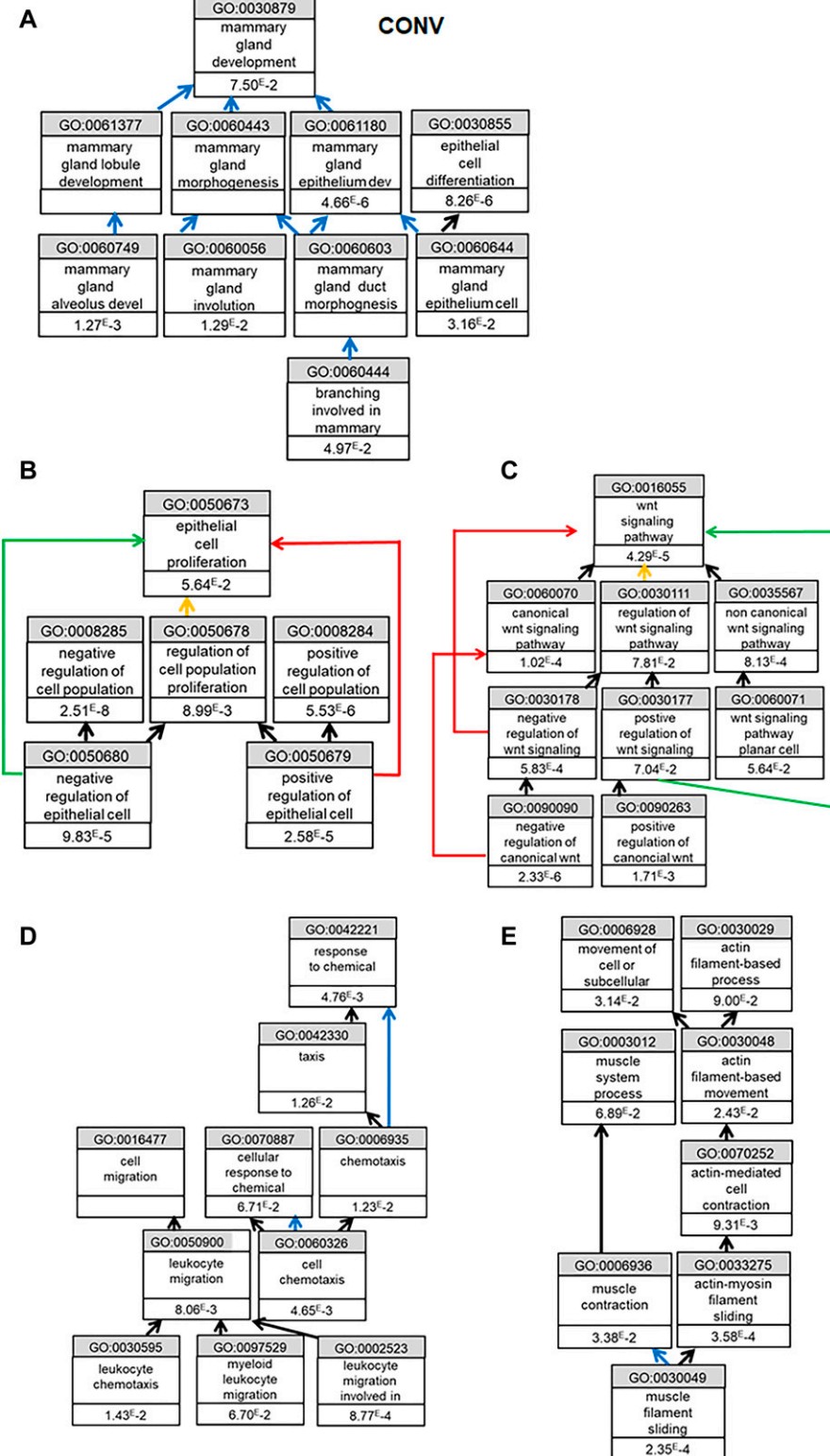

**Figure 5.** *Cxcr2* **KO affects mammary gland function in conventional conditions.**
**(A, B, C)** Gene Ontology analysis of down-regulated genes in the mammary gland of KO animals. Black arrows mean "is a." Blue arrows mean is "part of." Green arrow means "positively regulates." Red arrow means "negatively regulates." Yellow arrow means "regulates." **(A)** Biological process related to mammary gland function. **(B)** Similar analysis as in (B) in terms of cell proliferation. **(C)** Similar analysis as in (B) in terms of Wnt signaling. **(D, E)** Gene Ontology analysis of up-regulated genes in the mammary gland of KO animals. **(D)** Biological processes related to chemotaxis. **(E)** Biological processes related to muscle function.

the uteri of *Cxcr2 KO* animals housed conventionally, with global reductions in uterine size, and decreases in the endometrial thickness and the number of glands present, suggesting uterus in arrest not cycling under hormonal stimulation.

The mammary glands of *Cxcr2* KO females housed in conventional facilities were also minimally developed, harboring a phenotype close to that of juvenile animals with a rudimentary duct branching. In contrast, in SOPF conditions, mammary gland development appeared

**Table 4.  Most enriched pathways for genes up-regulated in the mammary gland of KO versus WT conventional animals.**

| GO biological process term | Count | % | *P*-value | Genes |
|---|---|---|---|---|
| GO:0030049~muscle filament sliding | 3 | 5.55 | 2.35E-4 | Myh6, Myh7, and Tnnc1 |
| GO:0055010~ventricular cardiac muscle tissue morphogenesis | 4 | 7.40 | 3.52E-4 | Myl3, Myh6, Myh7, and Tnnc1 |
| GO:0033275~actin–myosin filament sliding | 3 | 5.55 | 3.58E-4 | Myh6, Myh7, and Tnnc1 |
| GO:0003229~ventricular cardiac muscle tissue development | 4 | 7.40 | 5.07E-4 | Myl3, Myh6, Myh7, and Tnnc1 |
| GO:0055008~cardiac muscle tissue morphogenesis | 4 | 7.40 | 7.97E-4 | Myl3, Myh6, Myh7, and Tnnc1 |
| GO:0002523~leukocyte migration involved in inflammatory response | 3 | 5.55 | 8.77E-4 | S100a9, Ffar2, and S100a8 |

similar to that of WT animals. Interestingly, even in aged animals, no further development of the mammary gland could be observed in *Cxcr2* KO females housed in conventional facilities (data not shown). At the molecular level, the mammary gland of *Cxcr2* KO animals showed a clear down-regulation of the expression of genes involved in mammary gland development and differentiation, epithelial cell proliferation, wound healing, and Wnt signaling (including *lactoferrin, Gata3, cyclin D1, Erbb2, Elf5, Epcam, Prlr, Krt15, Epcam, Wnt4,* and *Wnt5b*). For a number of the down-regulated genes, invalidation studies in mice have shown that these genes (*Prlr, cyclin D1, Erbb2,* and *Elf5*) are crucial for mammary gland development (Bole-Feysot et al, 1998; Zhou et al, 2005; Howlin et al, 2006). The Wnt pathway plays a pivotal role in orchestrating proper mammary gland development and maintenance (Jarde & Dale, 2012), and this pathway was impaired in *Cxcr2* KO animals. The Wnt pathway is a major paracrine mediator of hormonal action, through Wnt4 and Areg pathways in particular (Brisken et al, 2000; Ciarloni et al, 2007), both of which are down-regulated in the mammary glands of *Cxcr2* KO animals housed conventionally. The general scheme of Wnt action in the mammary gland is based on the activation of estrogen receptor alpha (ERα)- positive luminal cells, which in turn are stimulated by steroid hormones to release Wnt ligands that will act directly or indirectly on the myoepithelial compartment (Jarde & Dale, 2012).

The collection of evidence gathered herein, for possible hormonal perturbations affecting *Cxcr2* KO animals housed in conventional conditions, including mammary, ovary, and uterus histologic changes and molecular defects, led us to evaluate steroid hormone levels in these animals. We observed, in particular, a decrease in serum progesterone levels, which could reflect the absence of corpora lutea in the ovary of these animals, the main site of production of progesterone.

To demonstrate the role of the hormonal environment in the phenotype observed for *Cxcr2* KO animals in conventional conditions, we decided to perform ovarian transplantation experiments. By transplanting the ovary of *Cxcr2* KO animals into the ovarian bursa of ovariectomized WT animals, we observed that KO ovaries could display a normal histology, close to WT ovaries, with the presence of corpora lutea, a sign of successive ovulations. Moreover, the transplanted animals had their fertility restored and were able to give birth to viable mice. Similarly, transplantation of the mammary glands of *Cxcr2* KO animals into a WT context also led to

the development of normal gland development with correct ductal branching. Taken together, this confirms that the hormonal context of WT animals is sufficient to restore a correct function and development of the *Cxcr2* KO ovary and mammary gland. As a function of the ovary, the uterus and the mammary gland are tightly controlled by steroid hormones, and at a higher level, by pituitary hormones, we measured the circulating levels of FSH, LH, GH, and PRL. The levels of these four pituitary hormones were markedly decreased in *Cxcr2* KO animals housed in conventional conditions compared with WT animals, suggesting an alteration to pituitary function. Treating the KO animals with FSH and LH to restore fertility could be interesting issue, but it is likely that the timing will be critical and difficult to assess because of the lack of cycling of these animals.

Transcriptomic and GO analysis of the pituitary of WT and *Cxcr2* KO animals revealed as expected, a down-regulation of genes involved in the control of circadian rhythm, ovulation control, and gonad development; which could account for dysregulation of pituitary hormones. This includes genes such as *Esr1* (estrogen receptor alpha), *Pgr* (progesterone receptor), *RMB4* (required for the translational activation of PER1 mRNA in response to circadian clock) (Markus & Morris, 2009), *Crebbp* (CREB-Binding Protein, involved in circadian clock) (Rexach et al, 2012), *Foxo3* (a regulator of circadian clock) (Chaves et al, 2014). Moreover, one could expect behavior alterations of Cxcr2 KO animals based on the down-regulation of a number of genes controlling behavior, including for instance *Crhbp* (corticotropin-releasing hormone binding protein) (Ketchesin et al, 2017), *Oprk1* (opioid receptor, kappa 1) (Loh et al, 2017), or *Vip* (vasoactive intestinal polypeptide) (Hill, 2007). The dysfunction of the *Cxcr2* KO pituitary involves presumably defects in synapse function, as well as ion transport. Indeed, we report a down-regulation of a number of genes involved in calcium, sodium, and potassium transport or in synaptic transmission such as *Vip, Oprk1, Kcnb2* (potassium voltage gated channel, *Shab*-related subfamily, member 2) or *Trpc6* (transient receptor potential cation channel, subfamily C, member 6), *Cacna1g* (calcium channel, voltage-dependent, T type, alpha 1G subunit). Synaptic alteration includes anterograde trans-synaptic signaling, long-term synaptic potentiation, and synaptic plasticity and could have major effects on neuronal connections. In addition to down-regulation of the pathways mentioned above, other pathways appeared up-regulated in the pituitary of *Cxcr2* KO animals. This includes in particular aggregation and adhesion of leukocytes as well as up-regulation of

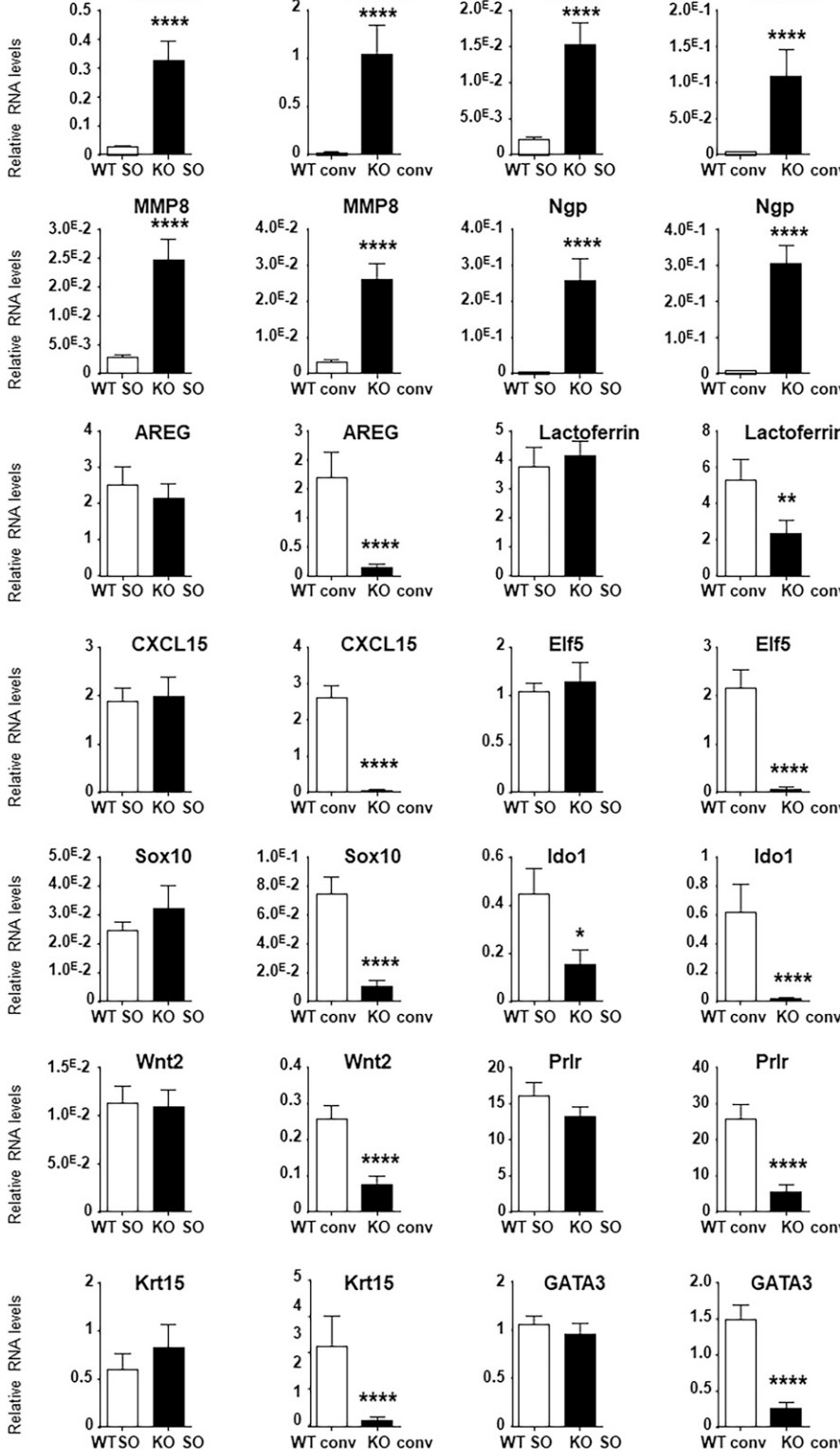

**Figure 6.  *Cxcr2* KO affects mammary gland transcriptome in conventional conditions.**
Measure of RNA levels by real-time PCR of a set of genes in the mammary gland of WT and KO animals in conventional or SOPF conditions. Results represent the mean the mean ± SEM of at least 12 animals (Mann–Whitney test, NS, nonsignificant, *P < 0.05, **P < 0.01, ***P < 0.001, ****P < 0.0001).

chemotaxis, extravasation, tethering, or rolling and also leukocyte activation, proliferation, and differentiation. When comparing our RNAseq results with signatures for B lymphocytes, T lymphocytes, macrophages, and neutrophils from Nirmal et al (2018), it appears

that there is a T lymphocyte enrichment (21%) and to a lesser extent of B lymphocytes (13.5%), macrophages (12.8%), and neutrophils (6.4%) (Table S1). This possible infiltration of T and B lymphocytes in the pituitary could be reminiscent of autoimmune hypophysitis

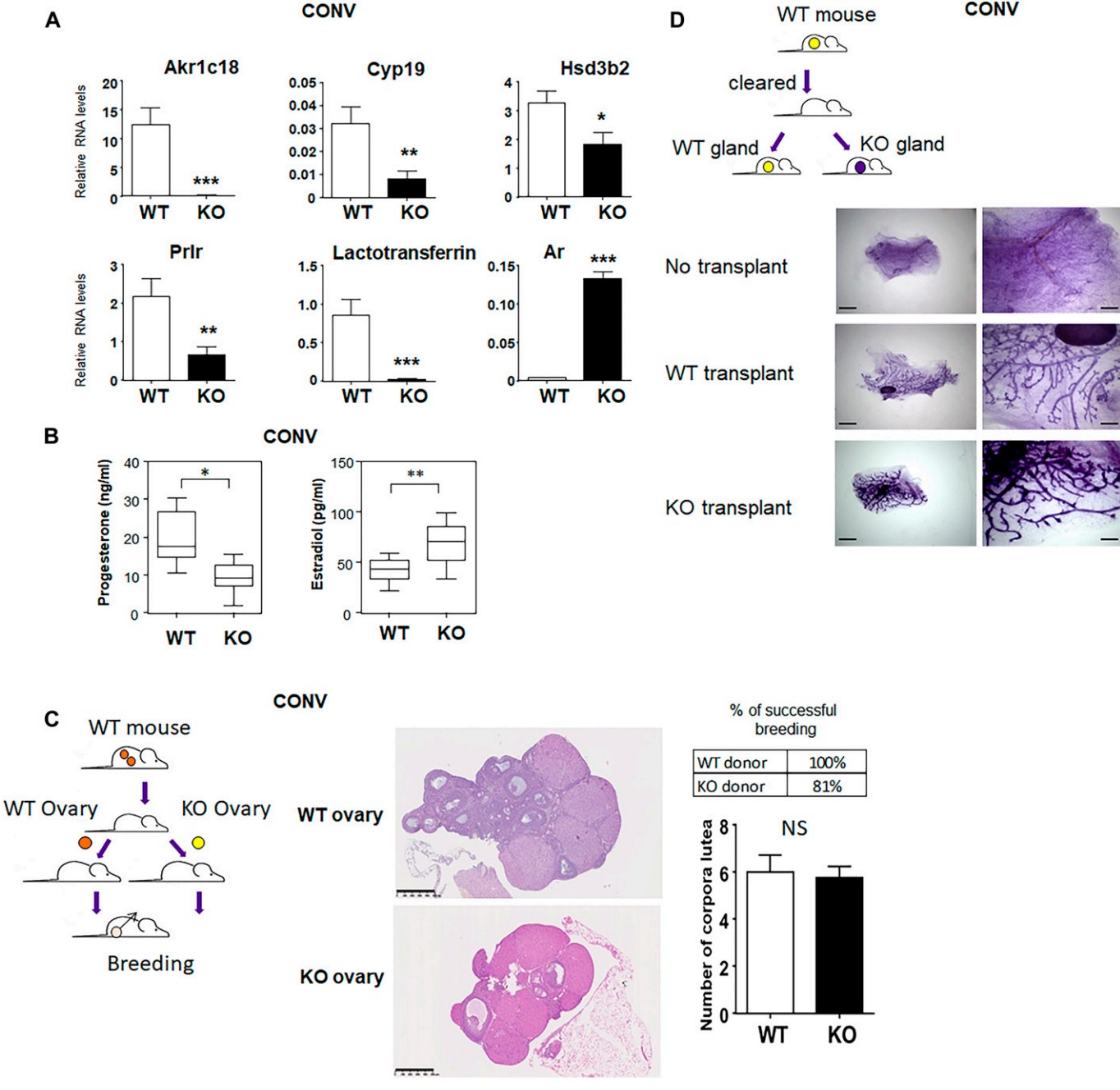

**Figure 7. *Cxcr2* KO display an alteration of hormonal function in conventional conditions.**
**(A)** Measure of RNA levels of a set of genes in the ovary of WT and KO animals in conventional conditions by real-time PCR. Results represent the mean the mean ± SEM of at least seven animals (Mann–Whitney test, NS, nonsignificant, *$P < 0.05$, **$P < 0.01$, ***$P < 0.001$). **(B)** Serum levels of estradiol and progesterone in WT and KO mice in conventional conditions. Box and whiskers represent the min and max of at least 10 animals (Mann–Whitney test, NS, nonsignificant, *$P < 0.05$, **$P < 0.01$). **(C)** Left panel: strategy of ovary transplantation. A *Cxcr2* WT mouse was ovariectomized and reimplanted with either WT or KO ovary from a conventional facility. Once the graft was established, females were bred with WT males to evaluate their fertility. Right panel: Histology of the transplanted *Cxcr2* WT and *Cxcr2* KO ovaries. Representative images of hematoxylin–eosin–stained ovaries at a 5× magnification are shown here. Scale bars: 500 μm. The % of successful breeding of transplanted recipient females is indicated. Fisher's exact test shows no difference between WT and KO successful breeding ($P = 0.5147$). The number of corpora lutea in WT or KO transplanted ovaries is also presented and shows no statistical difference (Mann–Whitney test, NS). **(D)** Left panel: strategy of mammary gland transplantation. The mammary gland fat pads of *Cxcr2* WT mice were cleared and transplanted with either *Cxcr2* WT or *Cxcr2* KO mammary gland. Right panel: Whole mounts of mammary glands of recipient mice after no transplantation or transplantation with WT or KO mammary glands. Scale bars: 5 mm (left panel) or 1.3 mm (right panel).

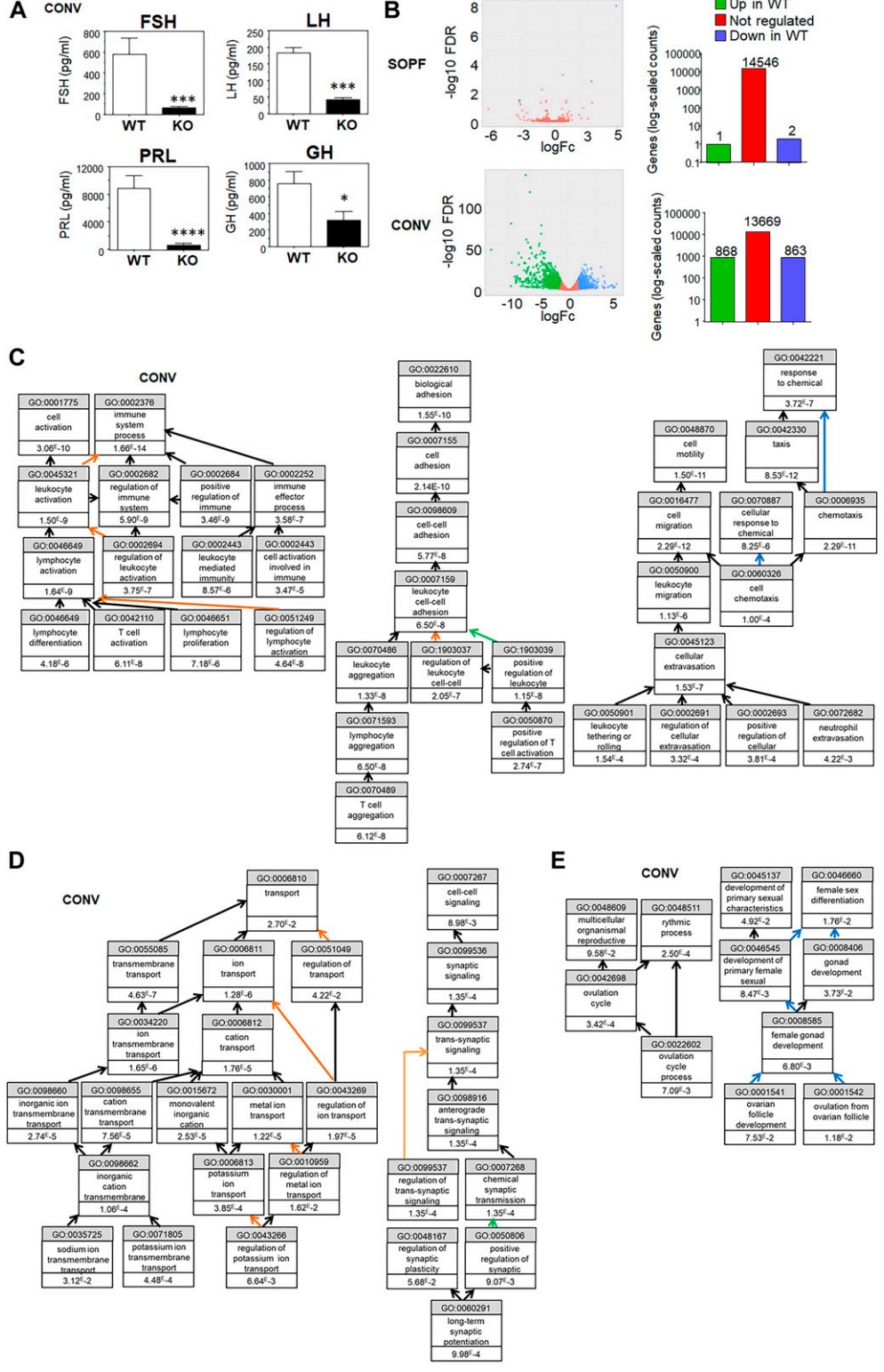

**Figure 8. Circulating pituitary hormones and transcriptome in the pituitary of KO animals are drastically affected in conventional housing conditions.**
**(A)** Serum levels of pituitary hormones FSH, LH, PRL, and GH. Results represent the mean ± SEM of at least 14 animals (Mann–Whitney test, NS, nonsignificant, *P < 0.05, **P < 0.01, ***P < 0.001, ****P < 0.0001). **(B)** Left panel: the volcano plots show the global changes in RNA expression patterns for WT versus KO pituitary in SOPF or conventional conditions. Data represent analysis of cpm estimates with a log of fold change of more than 1.5-fold and P < 0.05 of 3 animals per group. Right panel: Number of differentially regulated genes for the same analysis. **(C)** Gene Ontology (GO) analysis of up-regulated genes in the pituitary of KO animals in conventional conditions. **(D)** GO analysis of down-regulated genes in the pituitary of KO animals in conventional conditions linked to synapse function and ion transport. **(E)** GO analysis of down-regulated genes in the pituitary of *Cxcr2* KO animals in conventional conditions linked to reproduction.

(AH), which is an inflammatory disease of the pituitary gland, and particularly in relation to lymphocytic hypophysitis (Bellastella et al, 2016). AH can lead to atrophy of the pituitary and to hypopituitrism, which is in agreement with what we observed for *Cxcr2*

KO animals in conventional conditions. Functional disturbances observed in humans with AH include headaches, visual disturbances, hypothyroidism, hypogonadism, and an inability to lactate (Caturegli et al, 2005).

**Table 5. Most enriched pathways for genes up-regulated in the pituitary of KO versus WT conventional animals.**

| GO biological process term | Count | % | P-value | Genes |
|---|---|---|---|---|
| GO:0044707~single-multicellular organism process | 384 | 45.33 | 5.40E-29 | Cxcl1, Cdkn1c, Anks6, Rasip1, Elane, Hes7, Scel, Egr1, Trpv2, Ngef, Jag2, Ptger4, Adamtsl2, Etv4, E130012A19Rik, Spns2, Gfra4, H2-Ab1, Lyl1, Shb, Cd79a, Cxcl5, Nrxn2, Mmp15, Spn, Stk11, Hlx, Chadl, Il18r1, Clec9a, Arhgap4, Chil1, Flt3, Vax1, Casp1, Nr1h4, Chia1, Mmp9, Pitx1, Cbln1, Crlf2, Sox18, Junb, Dhx58, Dusp6, Nfam1, Vsx1, Gm11128, Pkdcc, Gpr35, Dnaic2, Trnp1, Nrtn, Inhbb, Sox1, Calca, Grin2d, Nkx2-2, Scx, Ccr2, Ctgf, Rnf207, Npas2, Fgfr3, Hpn, Cd3e, Ngfr, Cd3d, Tymp, Adm, Sema3b, Fzd1, Pcdh8, Icam1, Ucn3, Trim15, Cd40, Clec4d, Foxd1, Foxf2, Aatk, Cchcr1, Lfng, Chad, Apc2, Btk, Pllp, Efnb3, Pcsk2, Gfap, Napsa, Evpl, Lrg1, Alox12b, Dapk3, Itgam, Igsf9, Mapk13, Hic1, Sbno2, Dll3, Fst, Ltk, Col7a1, Tyro3, Shisa2, Pdgfa, Ltf, Mmp8, Plekhg5, Prrx2, Ltb, Ccdc88b, Dusp1, Hap1, Card9, Wif1, Unc93b1, Ankrd6, Smo, Tgm1, Cyp24a1, Id4, Sema3f, Mnx1, Wnt6, Pirb, Agrn, Jchain, Mir132, Nrbp2, S100a8, Colq, Col9a3, Sema6c, Scn1b, Gch1, Wnt5b, Ccdc64, Cebpa, Tnni3, Slc12a5, Stab2, Skor1, Tnfrsf25, Micall2, Hp, Pawr, Ccdc85c, Jak3, Hrh3, Col13a1, Il20ra, Pomc, Aipl1, Gas1, Sema3g, Ackr3, Olig1, Nxnl2, Trem3, Dpysl4, Dusp4, Nkx2-1, S100a9, Col9a1, Irx3, Nfatc4, Eln, Prtn3, Card11, Padi4, Chga, Spock1, Kif26a, Wfikkn2, Gpr68, Col19a1, Nptx2, Islr2, Pdlim3, Ncf1, Myh4, Npy, Gpc2, Sema4c, Foxc2, Myh14, Dusp5, Uty, Col23a1, Clcn2, Lrfn4, Nab2, Gata2, Map1s, Vav2, Zfpm1, Klf2, Gli1, Dll1, Ccl19, Svs2, Ptk7, Trp73, Lrrc38, Nlgn2, Tead3, Klf15, Nptx1, Ephb3, Cldn5, Sphk1, Clec5a, Il4ra, Hcls1, Safb2, Slc32a1, Msln, Ackr1, Zap70, Fgfr4, Maff, Esm1, Il27ra, Nr4a1, Lbh, Nefl, Tcf15, Kcnq4, Nrgn, Cables1, Tle6, Alk, Sox13, C3, Sema7a, Errfi1, St14, Foxo6, Zfp219, Spry1, Ptn, Icos, Fezf2, Pigr, Apoe, Ccr1, Gsdmd, Relb, Nkx3-2, Tbx2, Cebpd, Mdfi, Irf8, Dnm1, Cd300lf, Adrb2, Cd27, Cebpb, Sema5b, Cactin, Nfe2, Speg, Cbs, Tgfbi, Il3ra, Bmp6, Unc45b, Nptxr, Adam15, Camp, Isl2, Col2a1, Scnn1a, Grm2, Tbx18, Rgs14, Htra1, Ramp1, Lamb2, Mycl, Esrp2, Mir212, Vwa1, Casp4, F13a1, Ppp1r1b, Plk5, Gpr37l1, Sh2b2, Nexn, Wwc1, Ccl2, Atn1, Crocc, Prkg2, Sema6b, Ngb, Egr3, Tdrd9, Cdk5r2, Nkx2-4, Srpk3, Angptl4, Vtcn1, Rtn4rl2, Lrrc4b, Gadd45b, Pi16, Itga2b, Lox, Cspg5, Cep131, Arc, Smad6, Spr, Pcsk1n, Ngp, Pglyrp1, Prom1, Tbx1, Atoh8, Lhx2, Col18a1, Bmp2, Asb2, Fgr, Ebf4, Svs3a, Sfrp5, Vgf, Coro1a, Kcnk3, Cdh22, Rax, Dnaaf3, Nek8, Spo11, Gp1bb, Hapln3, Selp, Atp1a2, Metrn, Zc3h12a, Dusp2, Ascl1, Nck2, Id3, Fcgr2b, Cd1d1, Nell1, Sox17, Afap1l2, Sox2, Alox5, Runx3, Fjx1, Zic3, Adamts7, Fzd9, Hes6, Nrg2, Ltbp3, Slc35d3, Adra2a, Tsnaxip1, E4f1, Six2, Lingo1, Spi1, Mpo, Adgrb1, Dact3, Mmp14, P2ry2, Ephb6, Megf11, Svs3b, Col11a2, Tpbgl, Nr2f6, Kif7, and Zic2 |
| GO:0007275~multicellular organism development | 320 | 37.78 | 2.52E-21 | Cxcl1, Cdkn1c, Anks6, Rasip1, Hes7, Scel, Egr1, Trpv2, Ngef, Jag2, Ptger4, Adamtsl2, Etv4, E130012A19Rik, Spns2, H2-Ab1, Lyl1, Shb, Cd79a, Cxcl5, Nrxn2, Mmp15, Spn, Stk11, Hlx, Chadl, Il18r1, Arhgap4, Chil1, Flt3, Vax1, Nr1h4, Mmp9, Pitx1, Cbln1, Sox18, Junb, Dusp6, Nfam1, Vsx1, Gm11128, Pkdcc, Dnaic2, Trnp1, Nrtn, Inhbb, Sox1, Calca, Nkx2-2, Scx, Ccr2, Ctgf, Fgfr3, Hpn, Cd3e, Ngfr, Cd3d, Tymp, Adm, Sema3b, Fzd1, Pcdh8, Icam1, Clec4d, Cd40, Foxd1, Foxf2, Aatk, Cchcr1, Lfng, Chad, Apc2, Btk, Pllp, Efnb3, Pcsk2, Gfap, Evpl, Lrg1, Alox12b, Dapk3, Itgam, Igsf9, Hic1, Sbno2, Dll3, Fst, Ltk, Col7a1, Tyro3, Pdgfa, Shisa2, Ltf, Mmp8, Prrx2, Ltb, Hap1, Dusp1, Wif1, Ankrd6, Smo, Tgm1, Id4, Sema3f, Mnx1, Wnt6, Pirb, Agrn, Mir132, Nrbp2, S100a8, Colq, Col9a3, Sema6c, Scn1b, Wnt5b, Ccdc64, Cebpa, Tnni3, Slc12a5, Skor1, Tnfrsf25, Micall2, Hp, Pawr, Ccdc85c, Jak3, Col13a1, Gas1, Sema3g, Ackr3, Olig1, Dpysl4, Dusp4, Nkx2-1, S100a9, Col9a1, Irx3, Nfatc4, Eln, Prtn3, Card11, Spock1, Kif26a, Wfikkn2, Gpr68, Col19a1, Islr2, Pdlim3, Myh4, Npy, Gpc2, Sema4c, Foxc2, Myh14, Dusp5, Uty, Clcn2, Lrfn4, Nab2, Gata2, Map1s, Vav2, Zfpm1, Klf2, Gli1, Dll1, Ccl19, Ptk7, Trp73, Lrrc38, Nlgn2, Tead3, Klf15, Nptx1, Ephb3, Cldn5, Sphk1, Clec5a, Il4ra, Hcls1, Slc32a1, Safb2, Msln, Zap70, Fgfr4, Maff, Esm1, Il27ra, Nr4a1, Lbh, Nefl, Tcf15, Kcnq4, Nrgn, Cables1, Tle6, Sox13, Alk, C3, Sema7a, Errfi1, St14, Foxo6, Zfp219, Spry1, Icos, Ptn, Fezf2, Apoe, Ccr1, Relb, Tbx2, Nkx3-2, Cebpd, Mdfi, Irf8, Cd300lf, Adrb2, Cebpb, Cd27, Sema5b, Cactin, Nfe2, Speg, Tgfbi, Cbs, Il3ra, Bmp6, Unc45b, Nptxr, Camp, Adam15, Isl2, Col2a1, Tbx18, Rgs14, Htra1, Ramp1, Mycl, Esrp2, Lamb2, Mir212, Casp4, Plk5, Sh2b2, Nexn, Gpr37l1, Ccl2, Atn1, Sema6b, Ngb, Egr3, Tdrd9, Cdk5r2, Nkx2-4, Srpk3, Angptl4, Rtn4rl2, Lrrc4b, Gadd45b, Pi16, Lox, Arc, Cep131, Cspg5, Smad6, Spr, Prom1, Ngp, Pglyrp1, Tbx1, Atoh8, Lhx2, Col18a1, Bmp2, Asb2, Ebf4, Sfrp5, Vgf, Kcnk3, Cdh22, Rax, Dnaaf3, Nek8, Spo11, Hapln3, Metrn, Zc3h12a, Dusp2, Ascl1, Nck2, Id3, Cd1d1, Nell1, Sox17, Sox2, Runx3, Fjx1, Zic3, Adamts7, Fzd9, Nrg2, Hes6, Ltbp3, Tsnaxip1, E4f1, Six2, Lingo1, Spi1, Adgrb1, Dact3, Mmp14, P2ry2, Megf11, Col11a2, Tpbgl, Nr2f6, Kif7, and Zic2 |

| | | | | |
|---|---|---|---|---|
| GO:0048731~system development | 292 | 34.47 | 1.0E-20 | Cxcl1, Cdkn1c, Anks6, Rasip1, Hes7, Scel, Egr1, Trpv2, Ngef, Jag2, Ptger4, Adamtsl2, Etv4, E130012A19Rik, Spns2, H2-Ab1, Lyl1, Shb, Cd79a, Cxcl5, Nrxn2, Spn, Stk11, Hlx, Chadl, Il18r1, Arhgap4, Chil1, Flt3, Vax1, Nr1h4, Mmp9, Cbln1, Pitx1, Sox18, Junb, Nfam1, Vsx1, Pkdcc, Trnp1, Nrtn, Inhbb, Sox1, Nkx2-2, Ccr2, Scx, Ctgf, Fgfr3, Hpn, Cd3e, Ngfr, Cd3d, Tymp, Adm, Sema3b, Fzd1, Icam1, Clec4d, Cd40, Foxd1, Foxf2, Aatk, Lfng, Chad, Btk, Pllp, Efnb3, Pcsk2, Gfap, Evpl, Lrg1, Alox12b, Dapk3, Itgam, Igsf9, Sbno2, Dll3, Fst, Ltk, Tyro3, Pdgfa, Ltf, Prrx2, Ltb, Hap1, Ankrd6, Smo, Tgm1, Id4, Sema3f, Mnx1, Wnt6, Pirb, Agrn, Mir132, Nrbp2, S100a8, Colq, Col9a3, Sema6c, Scn1b, Wnt5b, Ccdc64, Cebpa, Tnni3, Slc12a5, Skor1, Micall2, Hp, Pawr, Ccdc85c, Jak3, Col13a1, Gas1, Sema3g, Ackr3, Olig1, Dpysl4, Nkx2-1, S100a9, Col9a1, Irx3, Nfatc4, Eln, Prtn3, Card11, Spock1, Kif26a, Wfikkn2, Gpr68, Col19a1, Islr2, Pdlim3, Myh4, Npy, Gpc2, Sema4c, Foxc2, Myh14, Uty, Clcn2, Lrfn4, Nab2, Gata2, Map1s, Vav2, Zfpm1, Klf2, Gli1, Dll1, Ccl19, Ptk7, Trp73, Lrrc38, Nlgn2, Tead3, Klf15, Nptx1, Ephb3, Cldn5, Sphk1, Clec5a, Il4ra, Hcls1, Slc32a1, Safb2, Msln, Zap70, Fgfr4, Maff, Esm1, Il27ra, Nr4a1, Nefl, Tcf15, Kcnq4, Nrgn, Cables1, Tle6, Sox13, Alk, C3, Sema7a, Errfi1, St14, Foxo6, Zfp219, Spry1, Icos, Ptn, Fezf2, Apoe, Ccr1, Relb, Tbx2, Nkx3-2, Cebpd, Mdfi, Irf8, Cd300lf, Adrb2, Cebpb, Cd27, Sema5b, Nfe2, Speg, Tgfbi, Cbs, Il3ra, Bmp6, Unc45b, Nptxr, Camp, Adam15, Isl2, Col2a1, Tbx18, Rgs14, Htra1, Ramp1, Mycl, Esrp2, Lamb2, Mir212, Casp4, Plk5, Sh2b2, Nexn, Gpr37l1, Ccl2, Atn1, Sema6b, Ngb, Egr3, Cdk5r2, Srpk3, Angptl4, Rtn4rl2, Lrrc4b, Pi16, Lox, Cspg5, Smad6, Spr, Prom1, Ngp, Pglyrp1, Tbx1, Atoh8, Lhx2, Col18a1, Bmp2, Asb2, Sfrp5, Vgf, Kcnk3, Cdh22, Rax, Dnaaf3, Nek8, Spo11, Hapln3, Metrn, Zc3h12a, Ascl1, Nck2, Id3, Cd1d1, Sox17, Nell1, Sox2, Runx3, Fjx1, Zic3, Adamts7, Fzd9, Nrg2, Hes6, Ltbp3, Six2, Lingo1, Spi1, Adgrb1, Dact3, Mmp14, P2ry2, Megf11, Col11a2, Tpbgl, Kif7, Nr2f6, and Zic2 |
| GO:0044767~single-organism developmental process | 344 | 40.61 | 1.45E-19 | Cxcl1, Cdkn1c, Anks6, Rasip1, Rbm38, Hes7, Scel, Egr1, Trpv2, Gfy, Ngef, Jag2, Ptger4, Adamtsl2, Etv4, E130012A19Rik, Spns2, H2-Ab1, Lyl1, Shb, Cd79a, Cxcl5, Nrxn2, Mmp15, Spn, Stk11, Hlx, Chadl, Il18r1, Arhgap4, Chil1, Flt3, Vax1, Casp1, Nr1h4, Mmp9, Pitx1, Cbln1, Sox18, Junb, Dusp6, Nfam1, Vsx1, Gm11128, Pkdcc, Dnaic2, Trnp1, Nrtn, Inhbb, Sox1, Calca, Nkx2-2, Scx, Ccr2, Ctgf, Fgfr3, Hpn, Cd3e, Ngfr, Cd3d, Tymp, Adm, Sema3b, Fzd1, Pcdh8, Icam1, Clec4d, Cd40, Foxd1, Foxf2, Aatk, Hck, Cchcr1, Lfng, Pcsk4, Chad, Apc2, Btk, Pllp, Igfbp2, Efnb3, Pcsk2, Gfap, Evpl, Lrg1, Alox12b, Dapk3, Itgam, Igsf9, Hic1, Sbno2, Dll3, Fst, Ltk, Col7a1, Tyro3, Pdgfa, Shisa2, Ltf, Mmp8, Prrx2, Ltb, Dusp1, Hap1, Wif1, Unc93b1, Ankrd6, Smo, Tgm1, Cfap73, Id4, Sema3f, Mnx1, Wnt6, Pirb, Agrn, Mir132, Nrbp2, S100a8, Colq, Col9a3, Itgb7, Sema6c, Scn1b, Wnt5b, Ccdc64, Cebpa, Tnni3, Slc12a5, Skor1, Tnfrsf25, Micall2, Hp, Pawr, Ccdc85c, Jak3, Col13a1, Gas1, Sema3g, Ackr3, Olig1, Dpysl4, Dusp4, Mmp25, Nkx2-1, S100a9, Col9a1, Irx3, Nfatc4, Eln, Prtn3, Card11, Padi4, Spock1, Kif26a, Wfikkn2, Gpr68, Col19a1, Islr2, Pdlim3, Myh4, Npy, Gpc2, Sema4c, Foxc2, Myh14, Dusp5, Uty, Tekt2, Cgn, Clcn2, Lrfn4, Nab2, Gata2, Map1s, Vav2, Zfpm1, Klf2, Gli1, Dll1, Ccl7, Ccl19, Ptk7, Svs2, Trp73, Lrrc38, Hmga1, Shroom1, Nlgn2, Tead3, Klf15, Nptx1, Ephb3, Cldn5, Sphk1, Clec5a, Il4ra, Hcls1, Safb2, Slc32a1, Msln, Zap70, Fgfr4, Maff, Esm1, Il27ra, Nr4a1, Lbh, Nefl, Tcf15, Kcnq4, Nrgn, Cables1, Tle6, Alk, Sox13, C3, Sema7a, Errfi1, St14, Foxo6, Zfp219, Spry1, Icos, Ptn, Fezf2, Apoe, Ccr1, Relb, Cfap53, Tbx2, Nkx3-2, Cebpd, Mdfi, Irf8, Cd300lf, Adrb2, Cebpb, Cd27, Sema5b, Cactin, Nfe2, Speg, Cbs, Tgfbi, Il3ra, Bmp6, Unc45b, Nptxr, Camp, Adam15, Isl2, Col2a1, Tbx18, Rgs14, Htra1, Ramp1, Lamb2, Mycl, Esrp2, Mir212, Casp4, Plk5, Sh2b2, Nexn, Gpr37l1, Wwc1, Ccl2, Atn1, Sema6b, Ngb, Egr3, Tdrd9, Cdk5r2, Nkx2-4, Srpk3, Angptl4, Rtn4rl2, Lrrc4b, Gadd45b, Pi16, Lox, Cspg5, Arc, Cep131, Smad6, Spr, Prom1, Ngp, Pglyrp1, Tbx1, Atoh8, Lhx2, Col18a1, Bmp2, Asb2, Fgr, Ebf4, Sfrp5, Vgf, Coro1a, Kcnk3, Cdh22, Rax, Dnaaf3, Nek8, Spo11, Fmnl1, Hapln3, Metrn, Zc3h12a, Dusp2, Ascl1, Nck2, Id3, Rhou, Cd1d1, Nell1, Sox17, Sox2, Runx3, Fjx1, Zic3, Adamts7, Fzd9, Hes6, Nrg2, Ltbp3, Tsnaxip1, E4f1, Six2, Lingo1, Spi1, Mpo, Adgrb1, Dact3, Mmp14, P2ry2, Megf11, Col11a2, Tpbgl, Nr2f6, Kif7, and Zic2 |

| | | | | |
|---|---|---|---|---|
| GO:0048856~anatomical structure development | 340 | 40.14 | 2.177E-19 | Cxcl1, Cdkn1c, Anks6, Rasip1, Rbm38, Hes7, Scel, Egr1, Trpv2, Gfy, Ngef, Jag2, Ptger4, Adamtsl2, Etv4, E130012A19Rik, Spns2, H2-Ab1, Lyl1, Shb, Cd79a, Cxcl5, Nrxn2, Mmp15, Spn, Stk11, Hlx, Chadl, Il18r1, Arhgap4, Chil1, Flt3, Vax1, Casp1, Nr1h4, Mmp9, Pitx1, Cbln1, Sox18, Junb, Dusp6, Nfam1, Vsx1, Gm11128, Pkdcc, Dnaic2, Trnp1, Nrtn, Inhbb, Sox1, Calca, Nkx2-2, Scx, Ccr2, Ctgf, Fgfr3, Hpn, Cd3e, Ngfr, Cd3d, Tymp, Adm, Sema3b, Fzd1, Pcdh8, Icam1, Clec4d, Cd40, Foxd1, Foxf2, Aatk, Hck, Cchcr1, Lfng, Pcsk4, Chad, Apc2, Btk, Pllp, Efnb3, Pcsk2, Gfap, Evpl, Lrg1, Alox12b, Dapk3, Itgam, Igsf9, Hic1, Sbno2, Dll3, Fst, Ltk, Col7a1, Tyro3, Pdgfa, Shisa2, Ltf, Mmp8, Prrx2, Ltb, Dusp1, Hap1, Wif1, Unc93b1, Ankrd6, Smo, Tgm1, Cfap73, Id4, Sema3f, Mnx1, Wnt6, Pirb, Agrn, Mir132, Nrbp2, S100a8, Colq, Col9a3, Itgb7, Sema6c, Scn1b, Wnt5b, Ccdc64, Cebpa, Tnni3, Slc12a5, Skor1, Tnfrsf25, Micall2, Hp, Pawr, Ccdc85c, Jak3, Col13a1, Gas1, Sema3g, Ackr3, Olig1, Dpysl4, Dusp4, Mmp25, Nkx2-1, S100a9, Col9a1, Irx3, Nfatc4, Eln, Prtn3, Card11, Spock1, Kif26a, Wfikkn2, Gpr68, Col19a1, Islr2, Pdlim3, Myh4, Npy, Gpc2, Sema4c, Foxc2, Myh14, Dusp5, Uty, Tekt2, Cgn, Clcn2, Lrfn4, Nab2, Gata2, Map1s, Vav2, Zfpm1, Klf2, Gli1, Dll1, Ccl7, Ccl19, Ptk7, Svs2, Trp73, Lrrc38, Shroom1, Nlgn2, Tead3, Klf15, Nptx1, Ephb3, Cldn5, Sphk1, Clec5a, Il4ra, Hcls1, Safb2, Slc32a1, Msln, Zap70, Fgfr4, Maff, Esm1, Il27ra, Nr4a1, Lbh, Nefl, Tcf15, Kcnq4, Nrgn, Cables1, Tle6, Alk, Sox13, C3, Sema7a, Errfi1, St14, Foxo6, Zfp219, Spry1, Icos, Ptn, Fezf2, Apoe, Ccr1, Relb, Cfap53, Tbx2, Nkx3-2, Cebpd, Mdfi, Irf8, Cd300lf, Adrb2, Cebpb, Cd27, Sema5b, Cactin, Nfe2, Speg, Cbs, Tgfbi, Il3ra, Bmp6, Unc45b, Nptxr, Camp, Adam15, Isl2, Col2a1, Tbx18, Rgs14, Htra1, Ramp1, Mycl, Esrp2, Lamb2, Mir212, Casp4, Plk5, Sh2b2, Nexn, Gpr37l1, Ccl2, Atn1, Sema6b, Ngb, Egr3, Tdrd9, Cdk5r2, Nkx2-4, Srpk3, Angptl4, Rtn4rl2, Lrrc4b, Gadd45b, Pi16, Lox, Cspg5, Arc, Cep131, Smad6, Spr, Prom1, Ngp, Pglyrp1, Tbx1, Atoh8, Lhx2, Col18a1, Bmp2, Asb2, Fgr, Ebf4, Sfrp5, Vgf, Coro1a, Kcnk3, Cdh22, Rax, Dnaaf3, Nek8, Spo11, Fmnl1, Hapln3, Metrn, Zc3h12a, Dusp2, Ascl1, Nck2, Id3, Rhou, Cd1d1, Nell1, Sox17, Sox2, Runx3, Fjx1, Zic3, Adamts7, Fzd9, Wtip, Hes6, Nrg2, Ltbp3, Tsnaxip1, E4f1, Six2, Lingo1, Spi1, Adgrb1, Dact3, Mmp14, P2ry2, Megf11, Col11a2, Tpbgl, Nr2f6, Kif7, and Zic2 |
| GO:0051239~regulation of multicellular organismal process | 202 | 23.84 | 2.79E-19 | Cxcl1, Cdkn1c, Elane, Hes7, Egr1, Trpv2, Ngef, Ptger4, Etv4, Gfra4, Shb, Cxcl5, Spn, Stk11, Hlx, Chadl, Il18r1, Clec9a, Flt3, Arhgap4, Chil1, Vax1, Casp1, Nr1h4, Chia1, Cbln1, Mmp9, Dusp6, Dhx58, Nfam1, Pkdcc, Gpr35, Inhbb, Calca, Grin2d, Nkx2-2, Ccr2, Scx, Ctgf, Rnf207, Fgfr3, Hpn, Cd3e, Ngfr, Adm, Sema3b, Fzd1, Icam1, Trim15, Foxd1, Cd40, Aatk, Lfng, Chad, Btk, Gfap, Lrg1, Alox12b, Mapk13, Fst, Dll3, Ltk, Pdgfa, Ltf, Ltb, Ccdc88b, Hap1, Card9, Unc93b1, Ankrd6, Smo, Id4, Sema3f, Wnt6, Agrn, Mir132, Colq, Sema6c, Scn1b, Cebpa, Tnni3, Stab2, Pawr, Jak3, Il20ra, Pomc, Sema3g, S100a9, Nkx2-1, Irx3, Nfatc4, Card11, Chga, Gpr68, Islr2, Ncf1, Sema4c, Foxc2, Gata2, Vav2, Gli1, Klf2, Zfpm1, Ccl19, Dll1, Ptk7, Trp73, Nlgn2, Tead3, Ephb3, Cldn5, Il4ra, Clec5a, Sphk1, Hcls1, Ackr1, Fgfr4, Zap70, Maff, Il27ra, Lbh, Nefl, Tle6, Sox13, C3, Sema7a, Errfi1, Foxo6, Spry1, Zfp219, Ptn, Fezf2, Ccr1, Apoe, Gsdmd, Relb, Tbx2, Nkx3-2, Cebpd, Irf8, Cebpb, Cd27, Adrb2, Sema5b, Nfe2, Cactin, Bmp6, Camp, Isl2, Tbx18, Rgs14, Mycl, Mir212, Casp4, Plk5, Gpr37l1, Wwc1, Ccl2, Sema6b, Egr3, Snta1, Vtcn1, Lrrc4b, Pi16, Spr, Smad6, Prom1, Pglyrp1, Ngp, Tbx1, Atoh8, Bmp2, Fgr, Sfrp5, Selp, Atp1a2, Metrn, Zc3h12a, Ascl1, Fcgr2b, Id3, Afap1l2, Sox17, Nell1, Cd1d1, Sox2, Alox5, Adamts7, Fzd9, Ltbp3, Adra2a, E4f1, Six2, Lingo1, Spi1, Adgrb1, Dact3, Mmp14, P2ry2, Tpbgl, and Zic2 |
| GO:0007166~cell surface receptor signaling pathway | 175 | 20.66 | 3.05E-19 | Cxcl1, Cdkn1c, Gpc2, Sema4c, Foxc2, Uty, Hes7, Dlk2, Egr1, Gata2, Ngef, Gli1, Jag2, Ccl19, Ccl7, Dll1, Ptk7, Adamtsl2, Grik5, Gfra4, Nlgn2, Shb, Cxcl5, Cd79a, Cd3g, Spn, Ephb3, Stk11, Cldn5, Il18r1, Lat2, Sphk1, Flt3, Ackr1, Lat, Zap70, Fgfr4, Esm1, Nr1h4, Rhbdf2, Mmp9, Alk, Sema7a, Errfi1, Nfam1, Spry1, Lcn2, Gpr35, Nrtn, Inhbb, Pigr, Ccr1, Styk1, Grin2d, Tbx2, Blk, Nkx2-2, Adgrg5, Fpr2, Ccr2, Scx, Ctgf, Mdfi, Fgfr3, Adrb2, Cd27, Sema5b, Cactin, Il3ra, Cd3e, Bmp6, Adam15, Ngfr, Cd3d, Sema3b, Col2a1, Fzd1, Tbx18, Rgs14, Icam1, Tle2, Foxd1, Cd40, Clec4d, Htra1, Hck, Matk, Gpr37l1, Sh2b2, Lfng, Chad, Apc2, Ccl2, Btk, Efnb3, Sema6b, Lrg1, Pmaip1, Dapk3, Rtn4rl2, Itgam, Itgax, Myo1g, Hic1, Itga2b, Ltbp4, Fst, Dll3, Arc, Ltk, Smad6, Shisa2, Pdgfa, Atoh8, Ltf, Ltb, Prrx2, Bmp2, Hap1, Fgr, Frat2, Epha10, Wif1, Ankrd6, Smo, Sfrp5, Tspan33, Coro1a, Igfbp4, Sema3f, Wnt6, Pirb, Podnl1, Ccl6, Ascl1, Nck2, Fcgr2b, Itgb7, Osmr, Afap1l2, Sox17, Sema6c, Pear1, Sox2, Wnt5b, Runx3, Cebpa, Skor1, Fzd9, Rhbdf1, Hp, Nrg2, Pawr, Ltbp3, Jak3, Adra2a, Il20ra, Mib2, Adgrb1, Gas1, Dact3, Sema3g, Mmp14, Ackr3, Csf2rb, Ephb6, Nkx2-1, Tpbgl, Nfatc4, Kif7, Card11, Pdzd3, Wfikkn2, and Zic2 |

| | | | | |
|---|---|---|---|---|
| GO:0032502~developmental process | 345 | 40.73 | 1.51E-18 | Cxcl1, Cdkn1c, Anks6, Rasip1, Rbm38, Hes7, Scel, Egr1, Trpv2, Gfy, Ngef, Jag2, Ptger4, Adamtsl2, Etv4, E130012A19Rik, Spns2, H2-Ab1, Lyl1, Shb, Cd79a, Cxcl5, Nrxn2, Mmp15, Spn, Stk11, Hlx, Chadl, Il18r1, Arhgap4, Chil1, Flt3, Vax1, Casp1, Nr1h4, Mmp9, Pitx1, Cbln1, Sox18, Junb, Dusp6, Nfam1, Vsx1, Gm11128, Pkdcc, Dnaic2, Trnp1, Nrtn, Inhbb, Sox1, Calca, Nkx2-2, Scx, Ccr2, Ctgf, Fgfr3, Hpn, Cd3e, Ngfr, Cd3d, Tymp, Adm, Sema3b, Fzd1, Pcdh8, Icam1, Clec4d, Cd40, Foxd1, Foxf2, Aatk, Hck, Cchcr1, Lfng, Pcsk4, Chad, Apc2, Btk, Pllp, Igfbp2, Efnb3, Pcsk2, Gfap, Evpl, Lrg1, Alox12b, Dapk3, Itgam, Igsf9, Hic1, Sbno2, Dll3, Fst, Ltk, Col7a1, Tyro3, Pdgfa, Shisa2, Ltf, Mmp8, Prrx2, Ltb, Dusp1, Hap1, Wif1, Unc93b1, Ankrd6, Smo, Tgm1, Cfap73, Id4, Sema3f, Mnx1, Wnt6, Pirb, Agrn, Mir132, Nrbp2, S100a8, Colq, Col9a3, Itgb7, Sema6c, Scn1b, Wnt5b, Ccdc64, Cebpa, Tnni3, Slc2a5, Skor1, Tnfrsf25, Micall2, Hp, Pawr, Ccdc85c, Jak3, Col13a1, Gas1, Sema3g, Ackr3, Olig1, Dpysl4, Dusp4, Mmp25, Nkx2-1, S100a9, Col9a1, Irx3, Nfatc4, Eln, Prtn3, Card11, Padi4, Spock1, Kif26a, Wfikkn2, Gpr68, Col19a1, Islr2, Pdlim3, Myh4, Npy, Gpc2, Sema4c, Foxc2, Myh14, Dusp5, Uty, Tekt2, Cgn, Clcn2, Lrfn4, Nab2, Gata2, Map1s, Vav2, Zfpm1, Klf2, Gli1, Dll1, Ccl7, Ccl19, Ptk7, Svs2, Trp73, Lrrc38, Hmga1, Shroom1, Nlgn2, Tead3, Klf15, Nptx1, Ephb3, Cldn5, Sphk1, Clec5a, Il4ra, Hcls1, Safb2, Slc32a1, Msln, Zap70, Fgfr4, Maff, Esm1, Il27ra, Nr4a1, Lbh, Nefl, Tcf15, Kcnq4, Nrgn, Cables1, Tle6, Alk, Sox13, C3, Sema7a, Errfi1, St14, Foxo6, Zfp219, Spry1, Icos, Ptn, Fezf2, Apoe, Ccr1, Relb, Cfap53, Tbx2, Nkx3-2, Cebpd, Mdfi, Irf8, Cd300lf, Adrb2, Cebpb, Cd27, Sema5b, Cactin, Nfe2, Speg, Cbs, Tgfbi, Il3ra, Bmp6, Unc45b, Nptxr, Camp, Adam15, Isl2, Col2a1, Tbx18, Rgs14, Htra1, Ramp1, Lamb2, Mycl, Esrp2, Mir212, Casp4, Plk5, Sh2b2, Nexn, Gpr37l1, Wwc1, Ccl2, Atn1, Sema6b, Ngb, Egr3, Tdrd9, Cdk5r2, Nkx2-4, Srpk3, Angptl4, Rtn4rl2, Lrrc4b, Gadd45b, Pi16, Lox, Cspg5, Arc, Cep131, Smad6, Spr, Prom1, Ngp, Pglyrp1, Tbx1, Atoh8, Lhx2, Col18a1, Bmp2, Asb2, Fgr, Ebf4, Sfrp5, Vgf, Coro1a, Kcnk3, Cdh22, Rax, Dnaaf3, Nek8, Spo11, Fmnl1, Hapln3, Metrn, Zc3h12a, Dusp2, Ascl1, Nck2, Id3, Rhou, Cd1d1, Nell1, Sox17, Sox2, Runx3, Fjx1, Zic3, Adamts7, Fzd9, Wtip, Hes6, Nrg2, Ltbp3, Tsnaxip1, E4f1, Six2, Lingo1, Spi1, Mpo, Adgrb1, Dact3, Mmp14, P2ry2, Megf11, Col11a2, Tpbgl, Nr2f6, Kif7, and Zic2 |
| GO:0051240~positive regulation of multicellular organismal process | 136 | 16.05 | 2.07E-18 | Cxcl1, Elane, Foxc2, Egr1, Trpv2, Gata2, Zfpm1, Gli1, Dll1, Ccl19, Ptger4, Ptk7, Trp73, Nlgn2, Tead3, Shb, Cxcl5, Spn, Ephb3, Stk11, Hlx, Il18r1, Il4ra, Clec5a, Sphk1, Hcls1, Clec9a, Flt3, Chil1, Zap70, Fgfr4, Casp1, Il27ra, Nefl, Nr1h4, Chia1, Tle6, Cbln1, Mmp9, Dhx58, C3, Sema7a, Nfam1, Foxo6, Spry1, Pkdcc, Zfp219, Ptn, Fezf2, Inhbb, Ccr1, Apoe, Gsdmd, Calca, Nkx2-2, Scx, Ccr2, Ctgf, Cebpd, Rnf207, Irf8, Fgfr3, Hpn, Adrb2, Cd27, Cebpb, Cd3e, Bmp6, Camp, Ngfr, Adm, Tbx18, Rgs14, Icam1, Trim15, Foxd1, Cd40, Casp4, Plk5, Gpr37l1, Ccl2, Gfap, Egr3, Lrg1, Alox12b, Vtcn1, Lrrc4b, Mapk13, Fst, Dll3, Ltk, Smad6, Pdgfa, Prom1, Atoh8, Tbx1, Ltf, Ltb, Ccdc88b, Bmp2, Hap1, Fgr, Card9, Unc93b1, Smo, Id4, Wnt6, Agrn, Selp, Metrn, Zc3h12a, Ascl1, Cd1d1, Nell1, Sox17, Afap1l2, Scn1b, Sox2, Alox5, Cebpa, Fzd9, Pawr, Ltbp3, Adra2a, Adgrb1, Mmp14, P2ry2, S100a9, Irx3, Nfatc4, Card11, Chga, Gpr68, Islr2, and Zic2 |
| GO:0009653~anatomical structure morphogenesis | 195 | 23.02 | 6.80E-18 | Myh4, Cdkn1c, Sema4c, Rasip1, Foxc2, Myh14, Dusp5, Uty, Hes7, Tekt2, Cgn, Lrfn4, Nab2, Trpv2, Gata2, Gfy, Ngef, Map1s, Vav2, Zfpm1, Klf2, Gli1, Jag2, Dll1, Ccl7, Ptger4, Ptk7, Trp73, Etv4, Lrrc38, Shroom1, Shb, Mmp15, Nptx1, Ephb3, Stk11, Hlx, Sphk1, Il4ra, Chil1, Arhgap4, Vax1, Fgfr4, Esm1, Casp1, Nr4a1, Nefl, Tcf15, Kcnq4, Mmp9, Pitx1, Cbln1, Sox18, Junb, Dusp6, C3, Sema7a, Errfi1, St14, Vsx1, Gm11128, Pkdcc, Zfp219, Spry1, Ptn, Dnaic2, Trnp1, Fezf2, Sox1, Apoe, Cfap53, Tbx2, Nkx3-2, Scx, Ccr2, Ctgf, Mdfi, Fgfr3, Hpn, Adrb2, Cebpb, Sema5b, Nfe2, Cbs, Tgfbi, Bmp6, Adam15, Camp, Ngfr, Tymp, Adm, Isl2, Col2a1, Sema3b, Fzd1, Tbx18, Pcdh8, Icam1, Foxd1, Htra1, Ramp1, Lamb2, Foxf2, Esrp2, Aatk, Hck, Lfng, Chad, Ccl2, Efnb3, Sema6b, Egr3, Cdk5r2, Lrg1, Dapk3, Angptl4, Sbno2, Dll3, Fst, Cep131, Arc, Smad6, Spr, Col7a1, Tyro3, Ngp, Pdgfa, Prom1, Tbx1, Atoh8, Ltf, Lhx2, Mmp8, Col18a1, Prrx2, Bmp2, Dusp1, Fgr, Hap1, Unc93b1, Ankrd6, Smo, Sfrp5, Tgm1, Coro1a, Cfap73, Id4, Sema3f, Dnaaf3, Nek8, Mnx1, Wnt6, Fmnl1, Agrn, Metrn, Zc3h12a, Dusp2, Id3, Rhou, Itgb7, Sox17, Sema6c, Scn1b, Sox2, Wnt5b, Runx3, Tnni3, Fjx1, Zic3, Wtip, Micall2, Pawr, Ltbp3, Six2, Col13a1, Lingo1, Spi1, Gas1, Adgrb1, Dact3, Mmp14, Sema3g, Ackr3, Dpysl4, Dusp4, Nkx2-1, Megf11, Col9a1, Irx3, Col11a2, Tpbgl, Nfatc4, Kif26a, Islr2, and Zic2 |

| | | | | |
|---|---|---|---|---|
| GO:0030154~cell differentiation | 261 | 30.81 | 1.14E-17 | Cdkn1c, Rasip1, Rbm38, Scel, Egr1, Dlk2, Trpv2, Ngef, Jag2, Ptger4, Etv4, E130012A19Rik, H2-Ab1, Lyl1, Shb, Cd79a, Cxcl5, Spn, Mmp15, Stk11, Hlx, Chadl, Il18r1, Arhgap4, Flt3, Vax1, Casp1, Cbln1, Pitx1, Mmp9, Junb, Sox18, Dusp6, Nfam1, Vsx1, Pkdcc, Nrtn, Inhbb, Sox1, Styk1, Calca, Myo7b, Blk, Nkx2-2, Ccr2, Scx, Ctgf, Fgfr3, Hpn, Cd3e, Ngfr, Cd3d, Adm, Sema3b, Fzd1, Icam1, Clec4d, Foxd1, Aatk, Hck, Cchcr1, Matk, Lfng, Pcsk4, Apc2, Btk, Efnb3, Gfap, Rasgrp4, Evpl, Lrg1, Dapk3, Igsf9, Itgam, Sbno2, Fst, Dll3, Ltk, Tyro3, Col7a1, Ltf, Mmp8, Hap1, Wif1, Smo, Tgm1, Cyp24a1, Id4, Sema3f, Mnx1, Wnt6, Pirb, Agrn, Mir132, Nrbp2, S100a8, Itgb7, Sema6c, Scn1b, Wnt5b, Ccdc64, Cebpa, Tnni3, Slc12a5, Skor1, Micall2, Pawr, Jak3, Col13a1, Gas1, Sema3g, Ackr3, Olig1, Dpysl4, S100a9, Nkx2-1, Irx3, Nfatc4, Prtn3, Card11, Spock1, Kif26a, Wfikkn2, Gpr68, Steap4, Col19a1, Islr2, Myh4, Npy, Gpc2, Sema4c, Foxc2, Clcn2, Cgn, Lrfn4, Nab2, Gata2, Map1s, Zfpm1, Klf2, Gli1, Ccl19, Dll1, Ptk7, Svs2, Trp73, Lrrc38, Nlgn2, Tead3, Klf15, Nptx1, Ephb3, Cldn5, Sphk1, Il4ra, Clec5a, Hcls1, Safb2, Zap70, Maff, Nr4a1, Nefl, Tcf15, Tle6, Sox13, Alk, Sema7a, Errfi1, St14, Foxo6, Zfp219, Ptn, Fezf2, Ccr1, Apoe, Relb, Tbx2, Nkx3-2, Trib3, Cebpd, Mdfi, Irf8, Cd300lf, Adrb2, Cebpb, Cd27, Sema5b, Speg, Tgfbi, Il3ra, Bmp6, Unc45b, Nptxr, Adam15, Isl2, Col2a1, Tbx18, Rgs14, Htra1, Mycl, Lamb2, Mir212, Casp4, Plk5, Nexn, Sh2b2, Gpr37l1, Ccl2, Sema6b, Ngb, Egr3, Tdrd9, Cdk5r2, Srpk3, Rtn4rl2, Gadd45b, Pi16, Cep131, Cspg5, Spr, Smad6, Prom1, Pglyrp1, Tbx1, Atoh8, Lhx2, Col18a1, Bmp2, Asb2, Fgr, Wfdc21, Sfrp5, Spo11, Metrn, Zc3h12a, Ascl1, Nck2, Id3, Cd1d1, Sox17, Nell1, Ccdc85b, Sox2, Runx3, Fkbp6, Zic3, Adamts7, Fzd9, Hes6, Ltbp3, Tsnaxip1, Six2, Lingo1, Spi1, Dact3, Mmp14, P2ry2, Tpbgl, Col11a2, Nr2f6, and Zic2 |
| GO:0048869~cellular developmental process | 275 | 32.46 | 1.77E-17 | Cdkn1c, Rasip1, Rbm38, Scel, Egr1, Dlk2, Trpv2, Gfy, Ngef, Jag2, Ptger4, Etv4, E130012A19Rik, H2-Ab1, Lyl1, Shb, Cd79a, Cxcl5, Spn, Mmp15, Stk11, Hlx, Chadl, Il18r1, Arhgap4, Flt3, Vax1, Casp1, Cbln1, Pitx1, Mmp9, Sox18, Junb, Dusp6, Nfam1, Vsx1, Pkdcc, Dnaic2, Nrtn, Inhbb, Sox1, Styk1, Calca, Myo7b, Blk, Nkx2-2, Ccr2, Scx, Ctgf, Fgfr3, Hpn, Cd3e, Ngfr, Cd3d, Adm, Sema3b, Fzd1, Icam1, Clec4d, Foxd1, Aatk, Hck, Cchcr1, Matk, Lfng, Pcsk4, Apc2, Btk, Efnb3, Gfap, Rasgrp4, Evpl, Lrg1, Dapk3, Igsf9, Itgam, Sbno2, Dll3, Fst, Ltk, Col7a1, Tyro3, Ltf, Mmp8, Hap1, Wif1, Unc93b1, Smo, Tgm1, Cfap73, Cyp24a1, Id4, Sema3f, Mnx1, Wnt6, Pirb, Agrn, Mir132, Nrbp2, S100a8, Itgb7, Sema6c, Scn1b, Wnt5b, Ccdc64, Cebpa, Tnni3, Slc12a5, Skor1, Micall2, Pawr, Jak3, Col13a1, Gas1, Sema3g, Ackr3, Olig1, Dpysl4, S100a9, Nkx2-1, Irx3, Nfatc4, Prtn3, Card11, Spock1, Kif26a, Wfikkn2, Gpr68, Steap4, Col19a1, Islr2, Myh4, Npy, Gpc2, Sema4c, Foxc2, Myh14, Clcn2, Cgn, Tekt2, Lrfn4, Nab2, Gata2, Map1s, Zfpm1, Klf2, Gli1, Dll1, Ccl19, Ccl7, Ptk7, Svs2, Trp73, Lrrc38, Hmga1, Shroom1, Nlgn2, Tead3, Klf15, Nptx1, Ephb3, Cldn5, Sphk1, Il4ra, Clec5a, Hcls1, Safb2, Zap70, Maff, Nr4a1, Nefl, Tcf15, Tle6, Sox13, Alk, Sema7a, Errfi1, St14, Foxo6, Zfp219, Ptn, Fezf2, Ccr1, Apoe, Relb, Cfap53, Tbx2, Nkx3-2, Trib3, Cebpd, Mdfi, Irf8, Cd300lf, Adrb2, Cebpb, Cd27, Sema5b, Speg, Tgfbi, Il3ra, Bmp6, Unc45b, Nptxr, Adam15, Isl2, Col2a1, Tbx18, Rgs14, Htra1, Mycl, Lamb2, Mir212, Casp4, Plk5, Nexn, Sh2b2, Gpr37l1, Ccl2, Sema6b, Ngb, Egr3, Tdrd9, Cdk5r2, Srpk3, Rtn4rl2, Gadd45b, Pi16, Cep131, Cspg5, Smad6, Spr, Prom1, Pglyrp1, Tbx1, Atoh8, Lhx2, Col18a1, Bmp2, Asb2, Fgr, Wfdc21, Sfrp5, Coro1a, Dnaaf3, Spo11, Fmnl1, Metrn, Zc3h12a, Ascl1, Nck2, Id3, Rhou, Cd1d1, Sox17, Nell1, Ccdc85b, Sox2, Runx3, Fkbp6, Zic3, Adamts7, Fzd9, Hes6, Ltbp3, Tsnaxip1, Six2, Lingo1, Spi1, Dact3, Mmp14, P2ry2, Tpbgl, Col11a2, Nr2f6, and Zic2 |
| GO:2000026~regulation of multicellular organismal development | 147 | 17.35 | 2.83E-16 | Cxcl1, Cdkn1c, Sema4c, Foxc2, Hes7, Egr1, Trpv2, Gata2, Ngef, Zfpm1, Gli1, Dll1, Ccl19, Ptger4, Ptk7, Trp73, Etv4, Nlgn2, Shb, Cxcl5, Ephb3, Stk11, Hlx, Chadl, Cldn5, Il4ra, Sphk1, Hcls1, Flt3, Chil1, Arhgap4, Vax1, Zap70, Fgfr4, Maff, Il27ra, Nefl, Nr1h4, Tle6, Cbln1, Mmp9, Sox13, Dusp6, C3, Sema7a, Errfi1, Nfam1, Foxo6, Pkdcc, Zfp219, Spry1, Ptn, Fezf2, Apoe, Ccr1, Nkx3-2, Tbx2, Nkx2-2, Scx, Ccr2, Ctgf, Fgfr3, Hpn, Adrb2, Cd27, Cebpb, Sema5b, Nfe2, Cd3e, Bmp6, Camp, Ngfr, Adm, Isl2, Sema3b, Fzd1, Tbx18, Rgs14, Foxd1, Cd40, Mycl, Aatk, Mir212, Plk5, Gpr37l1, Lfng, Chad, Ccl2, Sema6b, Gfap, Egr3, Lrg1, Lrrc4b, Pi16, Fst, Dll3, Ltk, Pglyrp1, Ngp, Pdgfa, Prom1, Atoh8, Tbx1, Ltf, Bmp2, Hap1, Ankrd6, Smo, Sfrp5, Id4, Sema3f, Wnt6, Agrn, Mir132, Metrn, Colq, Zc3h12a, Ascl1, Cd1d1, Nell1, Sox17, Sema6c, Scn1b, Sox2, Adamts7, Fzd9, Pawr, Ltbp3, Jak3, E4f1, Six2, Lingo1, Spi1, Adgrb1, Dact3, Sema3g, Mmp14, P2ry2, Nkx2-1, Irx3, Tpbgl, Nfatc4, Card11, Spock1, Gpr68, Islr2, and Zic2 |

<div align="right">(Continued on following page)</div>

| | | | |
|---|---|---|---|
| GO:0006928~movement of cell or subcellular component | 138 | 16.29 | 3.14E-16 |
| GO:0040011~locomotion | 127 | 14.99 | 4.88E-16 |
| GO:0050793~regulation of developmental process | 166 | 19.59 | 2.16E-15 |
| GO:0006954~inflammatory response | 70 | 8.26 | 2.28E-15 |
| GO:0006952~defense response | 121 | 14.28 | 8.90E-15 |
| GO:0002376~immune system process | 166 | 19.59 | 1.16E-14 |

**GO:0006928~movement of cell or subcellular component** — 138 — 16.29 — 3.14E-16

Cxcl1, Sema4c, Elane, Foxc2, Myh14, Uty, Tekt2, Egr1, Gata2, Vav2, Ccl7, Ccl19, Svs2, Ptger4, Ptk7, Etv4, Spns2, Kifc3, Cxcl5, Ephb3, Sphk1, Kiss1r, Arhgap4, Vax1, Fgfr4, Nr4a1, Nefl, Mmp9, Igsf8, Sox18, Sema7a, P2ry6, St14, Kif19a, Ptn, Gpr35, Dnaic2, Fezf2, Nrtn, Podxl2, Sox1, Apoe, Ccr1, Styk1, Calca, Cfap53, Selplg, Fpr2, Ccr2, Ctgf, Rnf207, Sema5b, Pstpip1, Adam15, Ngfr, Isl2, Sema3b, Icam1, Kif12, Foxd1, Lamb2, Matk, Nexn, Cldn7, Wwc1, Ccl2, Apc2, Kif21b, Efnb3, Atn1, Sema6b, Egr3, Cdk5r2, Snta1, Dapk3, Bin2, Itgam, Myo1g, Itga2b, Cep131, Arc, Tyro3, Pdgfa, Atoh8, Tbx1, Lhx2, Col18a1, Plekhg5, Bmp2, Hap1, Fgr, Klc3, Svs3a, Asap3, Smo, Coro1a, Cfap73, Amica1, Sema3f, Mnx1, Fmnl1, Agrn, Selp, Ccl6, Sell, S100a8, Atp1a2, Zc3h12a, Ascl1, Nck2, Itgb7, Sox17, Sema6c, Scn1b, Wnt5b, Runx3, Rhbdf1, Pawr, Jak3, Adra2a, Six2, Gas1, Sema3g, Mmp14, P2ry2, Trem3, Dpysl4, Saa3, Nkx2-1, S100a9, Retnlg, Tpbgl, Svs3b, Kif7, Spock1, Chga, Kif26a, and Zic2

**GO:0040011~locomotion** — 127 — 14.99 — 4.88E-16

Cxcl1, Sema4c, Elane, Foxc2, Tekt2, Egr1, Gata2, Vav2, Ccl7, Ccl19, Svs2, Ptger4, Ptk7, Etv4, Spns2, Nlgn2, Cxcl5, Ephb3, Sphk1, Arhgap4, Kiss1r, Vax1, Fgfr4, Nr4a1, Nefl, Mmp9, Sox18, Igsf8, Sema7a, P2ry6, St14, Ptn, Gpr35, Fezf2, Nrtn, Sox1, Podxl2, Apoe, Ccr1, Styk1, Calca, Selplg, Fpr2, Ccr2, Ctgf, Sema5b, Pstpip1, Adam15, Ngfr, Tymp, Isl2, Sema3b, Icam1, Foxd1, Lamb2, Nexn, Matk, Cldn7, Wwc1, Ccl2, Apc2, Efnb3, Atn1, Sema6b, Egr3, Cdk5r2, Dapk3, Bin2, Itgam, Myo1g, Itga2b, Arc, Tyro3, Pdgfa, Atoh8, Tbx1, Lhx2, Col18a1, Plekhg5, Bmp2, Fgr, Svs3a, Asap3, Smo, Coro1a, Amica1, Sema3f, Mnx1, Fmnl1, Agrn, Selp, Ccl6, Sell, S100a8, Atp1a2, Zc3h12a, Ascl1, Nck2, Itgb7, Sox17, Sema6c, Scn1b, Wnt5b, Runx3, Cxcr6, Rhbdf1, Pawr, Jak3, Adra2a, Six2, Gas1, Mmp14, Sema3g, Ackr3, P2ry2, Trem3, Dpysl4, Saa3, Nkx2-1, S100a9, Retnlg, Tpbgl, Svs3b, Spock1, Chga, Kif26a, and Zic2

**GO:0050793~regulation of developmental process** — 166 — 19.59 — 2.16E-15

Myh4, Cxcl1, Cdkn1c, Sema4c, Foxc2, Myh14, Rbm38, Hes7, Nab2, Egr1, Trpv2, Gata2, Ngef, Zfpm1, Gli1, Ccl19, Ccl7, Dll1, Ptger4, Ptk7, Trp73, Etv4, Hmga1, Nlgn2, Tead3, Shb, Cxcl5, Ephb3, Stk11, Hlx, Chadl, Cldn5, Il4ra, Sphk1, Hcls1, Flt3, Chil1, Arhgap4, Vax1, Zap70, Fgfr4, Maff, Il27ra, Nefl, Nr1h4, Tle6, Cbln1, Mmp9, Sox13, Dusp6, C3, Sema7a, Errfi1, Nfam1, Foxo6, Spry1, Pkdcc, Zfp219, Ptn, Fezf2, Ccr1, Apoe, Nkx3-2, Tbx2, Nkx2-2, Ccr2, Scx, Ctgf, Fgfr3, Hpn, Adrb2, Cd27, Cebpb, Sema5b, Nfe2, Cd3e, Bmp6, Camp, Ngfr, Adm, Isl2, Sema3b, Fzd1, Tbx18, Rgs14, Icam1, Foxd1, Cd40, Mycl, Aatk, Mir212, Hck, Plk5, Gpr37l1, Lfng, Chad, Wwc1, Ccl2, Sema6b, Gfap, Egr3, Lrg1, Dapk3, Lrrc4b, Pi16, Fst, Dll3, Arc, Ltk, Spr, Pglyrp1, Ngp, Pdgfa, Prom1, Atoh8, Tbx1, Ltf, Bmp2, Hap1, Fgr, Ankrd6, Smo, Sfrp5, Coro1a, Id4, Sema3f, Wnt6, Fmnl1, Agrn, Mir132, Metrn, Colq, Zc3h12a, Ascl1, Rhou, Id3, Cd1d1, Nell1, Sox17, Sema6c, Scn1b, Sox2, Adamts7, Fzd9, Wtip, Pawr, Ltbp3, Jak3, E4f1, Six2, Lingo1, Spi1, Adgrb1, Dact3, Sema3g, Mmp14, P2ry2, Nkx2-1, Irx3, Tpbgl, Nfatc4, Card11, Spock1, Gpr68, Islr2, and Zic2

**GO:0006954~inflammatory response** — 70 — 8.26 — 2.28E-15

Cxcl1, Ltb4r1, Elane, Il17d, Itgam, Ptger1, Ccl19, Ccl7, Ptger4, Trp73, Tyro3, Pglyrp1, Cxcl5, Spn, Bmp2, Sphk1, Il4ra, Chil1, Ackr1, Zap70, Lat, Igfbp4, Nr1h4, Tnfaip8l2, Chia1, Crlf2, Selp, Ccl6, S100a8, Orm2, C3, Sema7a, Nfkbiz, Zc3h12a, Fcgr2b, Afap1l2, Ccr1, Apoe, Gsdmd, Alox5, Calca, Relb, Siglece, Cxcr6, Chil3, Tnfrsf25, Chst2, Hp, Ccr2, Fpr2, Adra2a, Adrb2, Cebpb, Cd27, Serpina3n, Pstpip1, Bmp6, Ngfr, Icam1, Cd40, Mmp25, Ephb6, S100a9, Saa3, Casp4, Hck, C4b, Ccl2, Btk, and Ncf1

**GO:0006952~defense response** — 121 — 14.28 — 8.90E-15

Cxcl1, Elane, Ccl7, Ccl19, Ptger4, Trp73, H2-Ab1, Cxcl5, Spn, Clec5a, Sphk1, Il4ra, Ackr1, Chil1, Lat, Zap70, Casp1, Il27ra, Nr1h4, Chia1, Crlf2, Dhx58, C3, Sema7a, Lcn2, Apoe, Ccr1, Gsdmd, Styk1, Calca, Relb, Chil3, Blk, Chst2, Fpr2, Ccr2, Irf8, Cd27, Cebpb, Adrb2, Cactin, Pstpip1, Bmp6, Adam15, Camp, Adamts4, Ngfr, Adm, Ptprcap, Grm2, Icam1, Trim15, Cd40, Clec4d, Htra1, Casp4, Cfb, Ctsg, Hck, Plk5, Matk, Cldn7, Ccl2, Btk, Ltb4r1, 9530003J23Rik, Il17d, Pmaip1, Dapk3, Itgam, Ptger1, Itgax, Sbno2, Cspg5, Pglyrp1, Tyro3, Ngp, Ltf, Ccdc88b, Asb2, Bmp2, Fgr, Card9, Unc93b1, Slpi, Coro1a, Igfbp4, H2-Eb1, Tnfaip8l2, Selp, Jchain, Ccl6, Orm2, S100a8, Nfkbiz, Zc3h12a, Fcgr2b, Cd1d1, Afap1l2, Gch1, Alox5, Siglece, Cxcr6, Stab2, Tnfrsf25, Hp, Jak3, Adra2a, Serpina3n, Mpo, Ackr3, Trem3, Ephb6, Mmp25, Saa3, S100a9, C1ra, Padi4, Chga, C4b, and Ncf1

**GO:0002376~immune system process** — 166 — 19.59 — 1.16E-14

Sppl2b, Cxcl1, Cdkn1c, Elane, Egr1, Gata2, Zfpm1, Klf2, Jag2, Ccl19, Ccl7, Dll1, Ptger4, Spns2, H2-Ab1, Lyl1, Shb, Cd79a, Cxcl5, Spn, Ephb3, Stk11, Lrmp, Hlx, Il18r1, Lat2, Il4ra, Tbc1d10c, Clec5a, Hcls1, Flt3, Lat, Zap70, Casp1, Il27ra, Rab33a, Nr1h4, Chia1, Mmp9, Crlf2, Sox13, Junb, Dhx58, C3, Sema7a, Nfam1, Lcn2, Icos, Gpr35, Pigr, Ccr1, Podxl2, Gsdmd, Styk1, Relb, Calca, Selplg, C7, Nkx3-2, Blk, Fpr2, Ccr2, Cebpd, Irf8, Fgfr3, H2-Q10, Cd300lf, Cebpb, Cd27, Cactin, Pstpip1, Il3ra, Cd3e, Bmp6, Adam15, Camp, Ngfr, Cd3d, Adm, Icam1, Trim15, Cd40, Clec4d, Htra1, Ctsg, Cfb, Casp4, Hck, Matk, Sh2b2, Lfng, Cldn7, Rasal3, Ccl2, Btk, Igfbp2, Efnb3, Zc3h12d, Egr3, H2-Q1, Pla2g2f, Ctse, Pmaip1, Dapk3, Vtcn1, Itgam, Itgax, Myo1g, Itga2b, Sbno2, Fst, Cspg5, Smad6, Pglyrp1, Tyro3, Tbx1, Ltf, Ltb, Ccdc88b, Asb2, Fgr, Card9, Bst1, Unc93b1, Slpi, Coro1a, H2-Eb1, Amica1, Tnfaip8l2, Pirb, Selp, Jchain, Ccl6, Sell, S100a8, Zc3h12a, Nck2, Fcgr2b, Itgb7, Cd1d1, Gch1, Runx3, Cebpa, Tnfrsf25, Fzd9, Hp, Pawr, Jak3, Spi1, Mpo, Treml2, Mmp14, Ackr3, Trem3, Ephb6, S100a9, Retnlg, C1ra, Prtn3, Card11, Padi4, Kdm5d, Chga, C4b, Gpr68, and Ncf1

| | | | |
|---|---|---|---|
| GO:0001816~cytokine production | 67 | 7.91 | 1.02E-13 | Elane, Egr1, Vtcn1, Klf2, Zfpm1, Dll1, Ccl19, Mapk13, Ptger4, Pglyrp1, Ltf, Cxcl5, Spn, Ltb, Ccdc88b, Fgr, Card9, Il18r1, Sphk1, Il4ra, Unc93b1, Clec5a, Clec9a, Chil1, Ackr1, Flt3, Fgfr4, Casp1, Il27ra, Nr1h4, Chia1, Crlf2, Dhx58, C3, Sema7a, Errfi1, Zc3h12a, Nfam1, Fcgr2b, Cd1d1, Afap1l2, Inhbb, Gsdmd, Relb, Runx3, Ccr2, Ltbp3, Pawr, Irf8, Jak3, Adra2a, Cebpb, Cd27, Cactin, Cd3e, Pomc, P2ry2, Trem3, Trim15, Cd40, Ephb6, Casp4, Nfatc4, Card11, Chga, Ccl2, and Btk |

| GO:0044699~single-organism process | 626 | 73.90 | 1.38E-13 | Anks6, Slc26a10, Card10, Elane, Rbm38, Hes7, Scel, Dlk2, Stac, Egr1, Trpv2, Gfy, Ngef, Sult2b1, Arhgap9, Jag2, Ptger4, E130012A19Rik, Spns2, Gfra4, Icam4, Acot1, H2-Ab1, Nrxn2, Stk11, Hlx, Il18r1, Clec9a, Flt3, Vax1, Casp1, Arhgap30, Nr1h4, Kcne4, Rhbdf2, Chia1, Pitx1, Crlf2, Junb, Sox18, Dusp6, Eml2, P2ry6, Vsx1, Gm11128, Pkdcc, Gpr35, Dnaic2, Trnp1, Nrtn, Podxl2, Styk1, Calca, Myo7b, Grin2d, Selplg, Cyp27a1, Nkx2-2, Ctgf, Rnf207, Npas2, Fgfr3, Grasp, Baiap3, Cd3e, Cd3d, Adm, Pim3, Pate4, Cacng6, Foxf2, Aatk, Hck, Cchcr1, Lfng, Pcsk4, Slc4a3, Chad, Apc2, Igfbp2, Slco4a1, Pcsk2, Gfap, Napsa, Atp12a, Rasgrp4, Bin2, Rasl11a, Igsf9, Itgam, Ptger1, Ddah2, Itgax, Ltbp4, Msh5, Slc44a4, Pdgfa, Col7a1, Espn, Mmp8, Ltb, Prrx2, Plekhg5, Hap1, Slc6a9, Card9, Wif1, Ankrd6, Tgm1, Cyp24a1, Lmnb1, Wnt6, Pvrl4, Jchain, Pex6, Podnl1, Nrbp2, Col9a3, Padi1, Osmr, Phlda1, Sema6c, Scn1b, Vill, Skor1, Kcnk13, Rassf10, Rhbdf1, Pawr, Hrh3, Il20ra, Fbxo2, Aipl1, Gas1, Sema3g, Olig1, Slc16a11, Dpysl4, Col9a1, Eln, Padi4, Spock1, Gpr68, Steap4, Nptx2, Col19a1, Dusp5, Col23a1, Tekt2, Cgn, Nab2, Gli1, Ccl19, Svs2, Atp13a2, Gldc, Nlgn2, Tead3, Plcb3, Klf15, Aldh1l1, Cd3g, Srebf2, Nptx1, Ephb3, Col6a4, Cldn5, Tbc1d10c, Lat2, Msln, Fgfr4, Esm1, Il27ra, Lbh, Nefl, Galnt14, Tcf15, Kcnq4, Nrgn, Cables1, Tle6, Alk, Sox13, Slc22a17, C3, Errfi1, St14, Foxo6, Kif19a, Zfp219, Map3k6, Apoe, Naprt, Gsdmd, Relb, C7, Lgals7, Adgrg5, Fpr2, Trib3, Ddx3y, Cebpd, Mdfi, Cd300lf, Sema5b, Cactin, Nfe2, Cbs, Pstpip1, Il3ra, Bmp6, Adam15, Rhpn1, Fxyd6, Isl2, Col2a1, Tbx18, Rgs14, Tle2, Htra1, Mycl, Lamb2, Vwa1, Cfb, F13a1, Sh2b2, Nexn, Pdia2, Mafa, Exoc3l, Rgs11, Rasal3, Kif21b, Atn1, Crocc, Zc3h12d, Ltb4r1, Egr3, Tdrd9, Pla2g2f, Ano8, Srpk3, Pmaip1, Angptl4, Impa2, Gadd45b, Lrrc4b, Lox, Itga2b, Pi16, Cspg5, Cep131, Pcsk1n, Spr, Smad6, Ngp, Prom1, Tbx1, Asb2, Wfdc21, Dok3, Fgr, Ebf4, Epha10, Shisa7, Klc3, Asap3, Tcirg1, Tchh, Prph, Kcnk3, Gpr62, Rax, Nek8, Spo11, Metrn, Icam5, Doc2a, Nck2, Rhou, Gpr6, Cd1d1, Nell1, Afap1l2, Ccdc85b, Sox2, Pear1, Runx3, Fjx1, Shd, Adamts7, Zic3, Fkbp6, Slc6a14, Nrg2, Hes6, Ltbp3, Slc47a2, Slc35d3, Adra2a, Tsnaxip1, E4f1, Six2, Spi1, Adgrb1, Slc2a6, Dact3, P2ry2, Rhbg, Galnt12, Slfn2, Tmem132a, Saa3, Col11a2, Svs3b, Nr2f6, Kif7, Zic2, Cxcl1, Cdkn1c, Rab20, Rasip1, Tmem145, Adamtsl2, Tat, Etv4, Grik5, Kifc3, Pacsin3, Lyl1, Shb, Cd79a, Cxcl5, Mmp15, Spn, Lrmp, Chadl, Kcnh3, Tifab, Arhgap4, Apol9a, Chil1, Mgat5b, Piezo1, Rab33a, Mmp9, Cbln1, Otof, Dhx58, Nfam1, Fut4, Inhbb, Aldh8a1, Arrdc2, Khdc1a, Sox1, Blk, Scx, Ccr2, Hpn, Hmha1, Slc35f3, Ngfr, Tymp, Sema3b, Fignl2, Fzd1, Pcdh8, Ucn3, Icam1, Mfsd7a, Trim15, Foxd1, Cd40, Clec4d, Tha1, Matk, Btk, Pllp, Efnb3, Rasd2, Lrg1, Evpl, Alox12b, Dapk3, Cracr2b, Safb, Slc16a6, Hic1, Mapk13, Kank3, Slc7a10, Sbno2, Dll3, Fst, Ltk, Tyro3, Gda, Shisa2, Ltf, Ccdc88b, Plk3, Dusp1, Unc93b1, Smo, Tspan33, Pla2g4e, Igfbp4, Cfap73, Id4, Sema3f, Tnfaip8l2, Mnx1, Pirb, Agrn, Mir132, S100a8, Colq, Apitd1, Itgb7, Gch1, Wnt5b, Ccdc64, Cebpa, Tnni3, Siglece, Slc12a5, Stab2, Tnfrsf25, Tm7sf2, Micall2, Hp, Jak3, Ccdc85c, Col13a1, Pomc, Treml2, Ackr3, Csf2rb, Dohh, Trem3, Nxnl2, Dusp4, Mmp25, Cbarp, S100a9, Acap1, Nkx2-1, Irx3, Retnlg, Prtn3, Nfatc4, Card11, Chga, Pdzd3, Kif26a, Wfikkn2, Islr2, Pkmyt1, Gpr132, Pdlim3, Ncf1, Myh4, Gpc2, Npy, Sema4c, Foxc2, Atp2a3, Myh14, Uty, Clcn2, Lrfn4, Gm266, Gata2, Mical1, Arl4d, Map1s, Vav2, Fhod1, Klf2, Zfpm1, Gpr162, Dll1, Ccl7, Ptk7, Plpp4, Trp73, Lrrc38, Hmga1, Sh3bp1, Shroom1, Kcng4, Caskin1, Il4ra, Clec5a, Sphk1, Pdlim4, Hcls1, Galnt9, Slc32a1, Cadm4, Safb2, Pygl, Ackr1, Kiss1r, Lat, Zap70, Maff, Nr4a1, Prss12, Cnnm1, Igsf8, Sema7a, Col5a3, Lcn2, Spry1, Ptn, Icos, Fezf2, Pigr, Ccr1, Cfap53, Hoga1, Nkx3-2, Tbx2, Slc13a3, Irf8, Tjp3, Dnm1, Adrb2, Cd27, Cebpb, Speg, Tgfbi, Klf16, Nptxr, Unc45b, Camp, Arpc1b, Scnn1a, Grm2, Kif12, Ramp1, Esrp2, Mir212, Casp4, Plk5, Ppp1r1b, Gpr37l1, Slc16a3, Cldn7, Wwc1, Ccl2, Chtf18, Prkg2, Sema6b, Ngb, Cds1, Cyth4, Rasl10a, Nkx2-4, Cdk5r2, Snta1, B3gat1, Vtcn1, Rtn4rl2, Apobr, Hnrnpm, Myo1g, Arc, Pglyrp1, Atoh8, Lhx2, Gdpd3, Col18a1, Prodh, Bmp2, Ccdc68, Psd, Slc27a2, Coro6, Frat2, Bst1, Svs3a, Nat8l, Vgf, Sfrp5, Coro1a, Amica1, Cdh22, Dnaaf3, Fmnl1, Hapln3, Gp1bb, Selp, Ccl6, Sell, Atp1a2, Vstm2l, Zc3h12a, Dusp2, Ascl1, Fcgr2b, Id3, Sox17, Alox5, Cxcr6, Rhov, Fzd9, Wtip, Cyp2d22, B4galnt1, Lingo1, Mpo, Mib2, Hsd17b14, Mmp14, Asphd2, Gem, Ephb6, Megf11, Pnn, Tpbgl, Kdm5d, Plch2, and Gabrd |

**Table 5. Continued**

| | | | |
|---|---|---|---|
| GO:0009888~tissue development | 139 | 16.41 | 2.187987473633545E-13 |
| | | | Myh4, Cdkn1c, Sema4c, Rasip1, Foxc2, Myh14, Dusp5, Hes7, Cgn, Scel, Egr1, Gata2, Zfpm1, Gli1, Klf2, Jag2, Dll1, Ptk7, Adamtsl2, Trp73, Etv4, Klf15, Mmp15, Hlx, Chadl, Cldn5, Safb2, Vax1, Fgfr4, Maff, Nr4a1, Nr1h4, Tcf15, Mmp9, Pitx1, Junb, Sox18, Dusp6, Sema7a, Errfi1, St14, Pkdcc, Zfp219, Spry1, Ptn, Nrtn, Ccr1, Nkx3-2, Tbx2, Nkx2-2, Scx, Ctgf, Fgfr3, Hpn, Cebpb, Adrb2, Sema5b, Speg, Nfe2, Tgfbi, Cbs, Bmp6, Adam15, Ngfr, Adm, Col2a1, Sema3b, Fzd1, Tbx18, Pcdh8, Icam1, Foxd1, Esrp2, Mycl, Lamb2, Foxf2, Nexn, Lfng, Sema6b, Srpk3, Evpl, Pi16, Sbno2, Fst, Dll3, Arc, Smad6, Col7a1, Pdgfa, Prom1, Atoh8, Tbx1, Ltf, Lhx2, Mmp8, Col18a1, Prrx2, Asb2, Bmp2, Dusp1, Ankrd6, Smo, Tgm1, Sfrp5, Id4, Sema3f, Mnx1, Wnt6, Dusp2, Ascl1, Id3, Sox17, Nell1, Sema6c, Sox2, Wnt5b, Runx3, Tnni3, Zic3, Adamts7, Fzd9, Pawr, Ltbp3, Six2, Spi1, Gas1, Dact3, Mmp14, Sema3g, Dusp4, Nkx2-1, Col9a1, Irx3, Col11a2, Nfatc4, Eln, Col19a1, Pdlim3, and Zic2 |
| GO:0048583~regulation of response to stimulus | 209 | 24.67 | 3.267E-13 |
| | | | Cxcl1, Cdkn1c, Card10, Elane, Egr1, Dlk2, Ngef, Jag2, Ptger4, Adamtsl2, Spns2, H2-Ab1, Shb, Cxcl5, Cd79a, Spn, Stk11, Hlx, Il18r1, Chil1, Casp1, Nr1h4, Rhbdf2, Mmp9, Dusp6, Dhx58, Nfam1, Gpr35, Inhbb, Blk, Ccr2, Ctgf, Npas2, Fgfr3, Cd3e, Ngfr, Adm, Pim3, Sema3b, Fignl2, Fzd1, Icam1, Trim15, Foxd1, Clec4d, Cd40, Ctsg, Lfng, Chad, Apc2, Btk, Igfbp2, Rasgrp4, Rasd2, Lrg1, Alox12b, Dapk3, Hic1, Sbno2, Fst, Dll3, Shisa2, Pdgfa, Tyro3, Ltf, Prrx2, Plekhg5, Hap1, Dusp1, Card9, Unc93b1, Wif1, Ankrd6, Smo, Tspan33, Igfbp4, Sema3f, Tnfaip8l2, Lmnb1, Agrn, Podnl1, S100a8, Sema6c, Gch1, Wnt5b, Siglece, Stab2, Skor1, Tnfrsf25, Rhbdf1, Pawr, Jak3, Il20ra, Gas1, Sema3g, Ackr3, Dusp4, S100a9, Nkx2-1, Nfatc4, Card11, Chga, C4b, Kif26a, Gpr68, Sppl2b, Npy, Sema4c, Foxc2, Gata2, Vav2, Gli1, Ccl19, Ccl7, Dll1, Trp73, Hmga1, Nlgn2, Il4ra, Tbc1d10c, Sphk1, Lat2, Hcls1, Safb2, Kiss1r, Lat, Fgfr4, Zap70, Il27ra, Esm1, Lbh, Alk, C3, Sema7a, Errfi1, Spry1, Fezf2, Pigr, Map3k6, Ccr1, Apoe, Fpr2, Trib3, Mdfi, Dnm1, Cd27, Adrb2, Sema5b, Cactin, Cbs, Bmp6, Col2a1, Tbx18, Rgs14, Tle2, Htra1, Ramp1, Casp4, Cfb, Gpr37l1, Sh2b2, Pdia2, Rgs11, Cldn7, Rasal3, Wwc1, Ccl2, Sema6b, Ngb, Cyth4, Pmaip1, Rtn4rl2, Gadd45b, Lrrc4b, Myo1g, Pi16, Arc, Cspg5, Smad6, Pglyrp1, Tbx1, Bmp2, Asb2, Psd, Fgr, Sfrp5, Nek8, Selp, Sell, Ccl6, Zc3h12a, Ascl1, Nck2, Fcgr2b, Afap1l2, Sox17, Cd1d1, Pear1, Sox2, Runx3, Fzd9, Wtip, Adra2a, Mib2, Dact3, Mmp14, C1ra, Kif7, and Zic2 |

Whereas multiple links between gut microbiota and the hypothalamic–pituitary axis have been reported (Appleton, 2018), our results constitute the very first evidence for the involvement of a chemokine receptor, *Cxcr2*, in the control of pituitary function and the reproductive system and that this phenomenon is dependent on microbiota (Fig 10). Although there have been no reports of any action of Cxcr2 in pituitary function till date, Cxcr2 has been shown to be important for the positioning of oligodendrocyte precursors in the spinal cord (Tsai et al, 2002) and in myelin repair (Liu et al, 2010), but overall, the effects of Cxcr2 in central nervous system are poorly understood. On the other hand, a few reports have suggested a potential role for Cxcr2 in controlling the effects of microbiota. Indeed, it was shown that microbiota promotes the recruitment of Cxcr2-positive neutrophils that can protect from amebic colitis (Watanabe et al, 2017). Finally, so far, no direct evidence of a role of Cxcr2 in the control of reproduction has been reported. Some reports suggest that Cxcr2 might contribute to the development of preeclampsia (Wu et al, 2016; Chen et al, 2019) and in the control, pregnancy tolerance through neutrophils (Kang et al, 2016). In addition, loss of Cxcr2 ligand has been shown to result in premature senescence in placenta-derived mesenchymal stem cells (Li et al, 2017).

In summary, our results highlight a unique unexplored role for Cxcr2 in pituitary and reproductive function that is dependent on chronic infections (summarized in Fig 10). These findings reinforce an imperative; to be translationally relevant, the use of animal models to understand disease mechanisms must take into account animal husbandry conditions and disease phenotyping requiring the absence or presence of chronic infections.

# Materials and Methods

### Animal models and housing

All animal experiments conform to our animal protocols that were reviewed and approved by the Institutional Animal Care and Use Committee. *Cxcr2−/−* mice (Cacalano et al, 1994) obtained from the Jackson Laboratory were in BALB/c genetic background as well as their littermate control (WT) mice. Mice were either housed in an SOPF facility or transferred to a conventional animal facility. To favor the acquisition of microbiota from conventional facility to the SOPF animals which were transferred to conventional conditions, the bedding of conventional animals was added to one of the transferred SOPF animals during the first weeks. The phenotype of the animals housed in the conventional facility was first analyzed following at least one or two generations of breeding to enable the establishment of bystander infections in all animals. Because of the reduced fertility of *Cxcr2−/−* animals in conventional conditions, homozygous animals were obtained by crossing heterozygous animals.

### Pathogen screening

The presence of potential infections was monitored by Charles River Research Animal Diagnostic Services on whole animals or body swabs, oral swabs, and feces of the animals according to PCR rodent infectious agent (PRIA) instructions. The main bystander infections observed are shown in Fig S1.

**Table 6. Most enriched pathways for genes down-regulated in the pituitary of KO versus WT conventional animals.**

| GO biological process term | Count | % | P-value | Genes |
|---|---|---|---|---|
| GO:0007156~homophilic cell adhesion via plasma membrane adhesion molecules | 40 | 5.01 | 6.11E-23 | Pcdhga10, Pcdhga12, Pcdha11, Pcdhac2, Pcdha12, Pcdhb6, Pcdhb11, Pcdhgb1, Pcdhgc5, Pcdhb21, Pcdh10, Pcdhga9, Pcdhb8, Pcdha8, Pcdhga5, Pcdhga3, Pcdhgb8, Pcdhgb7, Pcdhgb5, Pcdhac1, Igsf9b, Pcdha9, Pcdhgb4, Pcdhgb2, Pcdhga8, Pcdhga6, Cdh12, Pcdhb19, Pcdhgb6, Cdh18, Pcdhb2, Pcdhb1, Pcdhb16, Pcdhga7, Sdk2, Cdh7, Pcdh11x, Pcdhga4, Pcdha5, and Pcdhga11 |
| GO:0098742~cell–cell adhesion via plasma membrane adhesion molecules | 42 | 5.26 | 9.39E-19 | Pcdhga10, Pcdhga12, Pcdha11, Pcdhac2, Pcdha12, Pcdhb6, Pcdhb11, Pcdhgb1, Pcdhgc5, Tenm1, Pcdhb21, Pcdh10, Pcdhga9, Pcdhb8, Pcdha8, Pcdhga5, Pcdhga3, Pcdhgb8, Pcdhgb7, Pcdhgb5, Pcdhac1, Igsf9b, Grid2, Pcdha9, Pcdhgb4, Pcdhgb2, Pcdhga8, Pcdhga6, Cdh12, Pcdhb19, Pcdhgb6, Cdh18, Pcdhb2, Pcdhb1, Pcdhb16, Pcdhga7, Sdk2, Cdh7, Pcdh11x, Pcdhga4, Pcdha5, and Pcdhga11 |
| GO:0045653~negative regulation of megakaryocyte differentiation | 9 | 1.12 | 2.35E-8 | Hist1h4a, Hist1h4c, Hist1h4i, Hist1h4f, Hist1h4j, Hist1h4d, Hist1h4h, Hist1h4k, and Hist2h4 |
| GO:0055085~transmembrane transport | 64 | 8.02 | 4.63E-7 | Slc36a4, G6pdx, Tspan13, Ucp1, Atp5g1, Lrrc8b, Slc14a1, Kcnv2, Utrn, Slc24a2, Atp7b, Slc17a9, Ryr2, Clcn5, Kcnk9, Cacna1c, Kcna6, Kcnd3, Slc1a7, Abcb1b, Sec61b, Gabrb2, Tmem245, Slc4a5, Asic2, Kcnip1, Kcna3, Nos1, Cacna1g, Crhbp, Slc30a7, Kcnb2, Trpc6, Htr1b, Gabra1, Slc8a1, Wnk3, Oprk1, Akap6, Mrs2, Slc30a8, Kcns2, Slc16a7, Gal, Kcnj6, Slc26a2, Atp6ap1l, Cacna1e, Kcnf1, Kcnh5, Tmem266, Aqp7, Slc5a8, Slc6a18, Nos1ap, Slc7a7, Slc5a7, Scnn1g, Reln, Klhl24, Oaz3, Slc8a3, Slc9a7, and Scn10a |
| GO:0006811~ion transport | 78 | 9.77 | 1.28E-6 | Slc36a4, G6pdx, Ano4, Vip, Ank1, Nkain3, Tspan13, Mif, Ucp1, Atp5g1, Slc15a1, Lrrc8b, Utrn, Kcnv2, Cd84, Grin2a, Slc24a2, Atp7b, Slc17a9, Ryr2, Clcn5, Kcnk9, Cacna1c, Kcna6, Kcnd3, Slc1a7, Ano5, Gabrb2, Slc4a5, Asic2, Kcnip1, Kcna3, Nos1, Cacna1g, Crhbp, Slc30a7, Kcnb2, Trpc6, Htr1b, Gabra1, Slc8a1, Wnk3, Slc41a2, Oprk1, Mrs2, Akap6, Slc30a8, Kcns2, Slc16a7, Kcnj6, Gal, Slc26a2, Atp6ap1l, Pla2r1, Klhl3, Cacna1e, Kcnf1, Grid2, Kcnh5, Steap3, Tmem266, Aqp7, Slc5a8, Slc6a18, Nos1ap, Slc7a7, Slc5a7, Scnn1g, Grin2b, Reln, Best3, Klhl24, Nkain2, Chrna9, Slc8a3, Slc9a7, Cckar, and Scn10a |
| GO:0034220~ion transmembrane transport | 48 | 6.01 | 1.65E-6 | Crhbp, Cacna1g, Kcnb2, Trpc6, Htr1b, G6pdx, Gabra1, Slc8a1, Tspan13, Wnk3, Oprk1, Atp5g1, Lrrc8b, Akap6, Kcnv2, Utrn, Slc24a2, Slc16a7, Kcns2, Atp7b, Ryr2, Gal, Kcnj6, Slc26a2, Clcn5, Atp6ap1l, Kcnk9, Cacna1c, Cacna1e, Kcnf1, Kcnh5, Kcna6, Aqp7, Kcnd3, Nos1ap, Slc1a7, Scnn1g, Reln, Gabrb2, Klhl24, Asic2, Slc4a5, Slc8a3, Kcnip1, Kcna3, Nos1, Slc9a7, and Scn10a |
| GO:0030001~metal ion transport | 50 | 6.26 | 1.22E-5 | Cacna1g, Slc30a7, Kcnb2, Trpc6, G6pdx, Htr1b, Vip, Slc8a1, Nkain3, Tspan13, Wnk3, Mif, Slc41a2, Oprk1, Akap6, Mrs2, Slc30a8, Utrn, Kcnv2, Grin2a, Cd84, Slc24a2, Kcns2, Atp7b, Kcnj6, Gal, Ryr2, Klhl3, Kcnk9, Cacna1c, Cacna1e, Kcnf1, Kcnh5, Steap3, Kcna6, Slc5a8, Kcnd3, Nos1ap, Slc5a7, Scnn1g, Grin2b, Nkain2, Asic2, Slc8a3, Kcnip1, Kcna3, Nos1, Slc9a7, Cckar, and Scn10a |
| GO:0006812~cation transport | 55 | 6.89 | 1.76E-5 | G6pdx, Vip, Ank1, Nkain3, Tspan13, Mif, Ucp1, Atp5g1, Kcnv2, Utrn, Cd84, Grin2a, Slc24a2, Atp7b, Ryr2, Kcnk9, Cacna1c, Kcna6, Kcnd3, Asic2, Kcnip1, Nos1, Kcna3, Cacna1g, Slc30a7, Kcnb2, Trpc6, Htr1b, Slc8a1, Slc41a2, Wnk3, Oprk1, Mrs2, Akap6, Slc30a8, Kcns2, Gal, Kcnj6, Klhl3, Atp6ap1l, Cacna1e, Kcnf1, Kcnh5, Steap3, Slc5a8, Nos1ap, Slc5a7, Scnn1g, Grin2b, Nkain2, Chrna9, Slc8a3, Slc9a7, Scn10a, and Cckar |

**Table 6.   Continued**

| GO term | Count | % | P-value | Genes |
|---|---|---|---|---|
| GO:0043269~regulation of ion transport | 40 | 5.01 | 1.97E-5 | Crhbp, Cacna1g, Kcnb2, Trpc6, G6pdx, Vip, Slc8a1, Nkain3, Tspan13, Wnk3, Mif, Oprk1, Slc15a1, Akap6, Utrn, Kcnv2, Cd84, Kcns2, Ryr2, Gal, Kcnj6, Pla2r1, Cacna1c, Cacna1e, Kcnf1, Kcnh5, Kcna6, Kcnd3, Nos1ap, Grin2b, Reln, Best3, Nkain2, Klhl24, Asic2, Kcnip1, Kcna3, Nos1, Scn10a, Cckar |
| GO:0015672~monovalent inorganic cation transport | 32 | 4.01 | 2.53E-5 | Kcnb2, Vip, Ank1, Slc8a1, Wnk3, Mif, Ucp1, Atp5g1, Oprk1, Akap6, Utrn, Kcnv2, Kcns2, Kcnj6, Gal, Klhl3, Atp6ap1l, Kcnk9, Kcnf1, Kcnh5, Kcna6, Slc5a8, Kcnd3, Slc5a7, Scnn1g, Asic2, Slc8a3, Kcnip1, Kcna3, Nos1, Slc9a7, and Scn10a |
| GO:0098660~inorganic ion transmembrane transport | 38 | 4.76 | 2.74E-5 | Cacna1g, Kcnb2, Trpc6, Htr1b, G6pdx, Gabra1, Slc8a1, Wnk3, Atp5g1, Oprk1, Akap6, Utrn, Kcnv2, Slc24a2, Kcns2, Atp7b, Ryr2, Kcnj6, Gal, Slc26a2, Clcn5, Atp6ap1l, Kcnk9, Cacna1c, Cacna1e, Kcnf1, Kcnh5, Kcna6, Kcnd3, Nos1ap, Scnn1g, Gabrb2, Slc8a3, Kcnip1, Kcna3, Nos1, Slc9a7, and Scn10a |
| GO:0007155~cell adhesion | 81 | 10.15 | 5.98E-5 | Pcdha11, Hist1h3e, Il2ra, Lamb3, Lpp, Itga1, Crisp2, Pcdhb11, Cntnap5c, Pcdhgc5, Utrn, Cd84, Tenm1, Pcdhb21, Pcdhga9, Tnr, Pcdha8, Pcdhga3, Nrxn3, Pcdhgb5, Igsf9b, Pcdha9, Pcdhgb4, Adam12, Cyfip2, Pkhd1, Hapln4, Pcdhb2, Edil3, Pcdhb16, Prkdc, Pcdhga7, Cdh7, Pcdh11x, Snai2, Agr2, Pcdhga4, Stat5a, Pcdhga10, Pcdhga12, Ncam2, Pcdhac2, Pcdha12, Pcdhb6, Omg, Pcdhgb1, Plet1, Scgb1a1, Omd, Cgref1, Crebbp, Pcdh10, Itgb8, Pcdhb8, Rps26, Pcdhga5, Pcdhgb8, Pcdhgb7, Sned1, Pcdhac1, Grid2, Gpam, Pcdhgb2, Pcdhga8, Epha5, Cntnap5b, Pcdhb19, Cdh12, Pcdhgb6, Pcdhga6, Cdh18, Hapln1, Pcdhb1, Tcam1, Reln, Sdk2, Phldb2, Flrt1, Pcdha5, S100a10, and Pcdhga11 |
| GO:0034762~regulation of transmembrane transport | 30 | 3.75 | 6.62E-5 | Crhbp, Kcnb2, Trpc6, G6pdx, Slc8a1, Tspan13, Wnk3, Oprk1, Akap6, Utrn, Kcnv2, Kcns2, Kcnj6, Gal, Ryr2, Cacna1c, Cacna1e, Kcnf1, Kcnh5, Kcna6, Kcnd3, Nos1ap, Reln, Klhl24, Asic2, Oaz3, Kcnip1, Nos1, Kcna3, and Scn10a |
| GO:0098655~cation transmembrane transport | 36 | 4.51 | 7.31E-5 | Cacna1g, Kcnb2, Trpc6, Htr1b, G6pdx, Slc8a1, Tspan13, Wnk3, Atp5g1, Oprk1, Akap6, Utrn, Kcnv2, Slc24a2, Kcns2, Atp7b, Ryr2, Kcnj6, Gal, Atp6ap1l, Kcnk9, Cacna1c, Cacna1e, Kcnf1, Kcnh5, Kcna6, Kcnd3, Nos1ap, Scnn1g, Asic2, Slc8a3, Kcnip1, Kcna3, Nos1, Slc9a7, and Scn10a |
| GO:0022610~biological adhesion | 81 | 10.15 | 7.56E-5 | Pcdha11, Hist1h3e, Il2ra, Lamb3, Lpp, Itga1, Crisp2, Pcdhb11, Cntnap5c, Pcdhgc5, Utrn, Cd84, Tenm1, Pcdhb21, Pcdhga9, Tnr, Pcdha8, Pcdhga3, Nrxn3, Pcdhgb5, Igsf9b, Pcdha9, Pcdhgb4, Adam12, Cyfip2, Pkhd1, Hapln4, Pcdhb2, Edil3, Pcdhb16, Prkdc, Pcdhga7, Cdh7, Pcdh11x, Snai2, Agr2, Pcdhga4, Stat5a, Pcdhga10, Pcdhga12, Ncam2, Pcdhac2, Pcdha12, Pcdhb6, Omg, Pcdhgb1, Plet1, Scgb1a1, Omd, Cgref1, Crebbp, Pcdh10, Itgb8, Pcdhb8, Rps26, Pcdhga5, Pcdhgb8, Pcdhgb7, Sned1, Pcdhac1, Grid2, Gpam, Pcdhgb2, Pcdhga8, Epha5, Cntnap5b, Pcdhb19, Cdh12, Pcdhgb6, Pcdhga6, Cdh18, Hapln1, Pcdhb1, Tcam1, Reln, Sdk2, Phldb2, Flrt1, Pcdha5, S100a10, and Pcdhga11 |
| GO:0034765~regulation of ion transmembrane transport | 29 | 3.63 | 7.69E-5 | Crhbp, Kcnb2, Trpc6, G6pdx, Slc8a1, Tspan13, Wnk3, Oprk1, Akap6, Utrn, Kcnv2, Kcns2, Kcnj6, Gal, Ryr2, Cacna1c, Cacna1e, Kcnf1, Kcnh5, Kcna6, Kcnd3, Nos1ap, Reln, Klhl24, Asic2, Kcnip1, Nos1, Kcna3, and Scn10a |
| GO:0098662~inorganic cation transmembrane transport | 34 | 4.26 | 1.06E-4 | Cacna1g, Kcnb2, Trpc6, Htr1b, G6pdx, Slc8a1, Wnk3, Atp5g1, Oprk1, Akap6, Utrn, Kcnv2, Slc24a2, Kcns2, Atp7b, Ryr2, Kcnj6, Gal, Atp6ap1l, Kcnk9, Cacna1c, Cacna1e, Kcnf1, Kcnh5, Kcna6, Kcnd3, Nos1ap, Scnn1g, Slc8a3, Kcnip1, Kcna3, Nos1, Slc9a7, and Scn10a |
| GO:0007268~chemical synaptic transmission | 37 | 4.63 | 1.35E-4 | Crhbp, Cacna1g, Ptpn5, Htr1b, Gabra1, Sytl5, Ssh1, Lrrtm2, Oprk1, Grin2a, Slc24a2, Cartpt, Cplx1, Syt6, Sstr2, Tnr, Rims4, Cacna1c, Nrxn3, Tmod2, Pclo, Igsf9b, Grid2, Gpr149, Slc5a7, Grin2b, Pcdhb16, Reln, Gabrb2, Chrna9, Gpr21, Adnp, Slc8a3, Slitrk5, Grm5, Nos1, and Retn |

| | | | | |
|---|---|---|---|---|
| GO:0099536~synaptic signaling | 37 | 4.63 | 1.35E-4 | Crhbp, Cacna1g, Ptpn5, Htr1b, Gabra1, Sytl5, Ssh1, Lrrtm2, Oprk1, Grin2a, Slc24a2, Cartpt, Cplx1, Syt6, Sstr2, Tnr, Rims4, Cacna1c, Nrxn3, Tmod2, Pclo, Igsf9b, Grid2, Gpr149, Slc5a7, Grin2b, Pcdhb16, Reln, Gabrb2, Chrna9, Gpr21, Adnp, Slc8a3, Slitrk5, Grm5, Nos1, and Retn |
| GO:0099537~trans-synaptic signaling | 37 | 4.63 | 1.35E-4 | Crhbp, Cacna1g, Ptpn5, Htr1b, Gabra1, Sytl5, Ssh1, Lrrtm2, Oprk1, Grin2a, Slc24a2, Cartpt, Cplx1, Syt6, Sstr2, Tnr, Rims4, Cacna1c, Nrxn3, Tmod2, Pclo, Igsf9b, Grid2, Gpr149, Slc5a7, Grin2b, Pcdhb16, Reln, Gabrb2, Chrna9, Gpr21, Adnp, Slc8a3, Slitrk5, Grm5, Nos1, and Retn |
| GO:0098916~anterograde trans-synaptic signaling | 37 | 4.63 | 1.35E-4 | Crhbp, Cacna1g, Ptpn5, Htr1b, Gabra1, Sytl5, Ssh1, Lrrtm2, Oprk1, Grin2a, Slc24a2, Cartpt, Cplx1, Syt6, Sstr2, Tnr, Rims4, Cacna1c, Nrxn3, Tmod2, Pclo, Igsf9b, Grid2, Gpr149, Slc5a7, Grin2b, Pcdhb16, Reln, Gabrb2, Chrna9, Gpr21, Adnp, Slc8a3, Slitrk5, Grm5, Nos1, and Retn |
| GO:0048511~rhythmic process | 27 | 3.38 | 2.50E-4 | Prok2, Foxo3, Setx, Oprk1, Pgr, Grin2a, Cartpt, Crebbp, Nhlh2, Dmrta1, Prox1, Gpr149, Nrip1, Esr1, Arntl2, Atm, Prl, Grin2b, Prkdc, Tyms, Rbm4, Alb, Uba52, Adnp, Retn, Mmp19, and Stat5a |
| GO:0042698~ovulation cycle | 14 | 1.75 | 3.42E-4 | Foxo3, Gpr149, Nrip1, Esr1, Atm, Prl, Oprk1, Pgr, Adnp, Nhlh2, Stat5a, Dmrta1, Mmp19, Retn |
| GO:0006813~potassium ion transport | 18 | 2.25 | 3.85E-4 | Kcnf1, Kcnh5, Kcnb2, Kcna6, Vip, Kcnd3, Mif, Oprk1, Akap6, Kcnv2, Kcns2, Gal, Kcnj6, Kcnip1, Kcnk9, Nos1, Kcna3, and Slc9a7 |
| GO:0065008~regulation of biological quality | 145 | 18.17 | 4.09E-4 | Slitrk3, Ano4, Foxo3, Vps13c, Vip, Frmpd4, Dock4, Lrrtm2, Mif, Fgd4, Ryr2, Tnr, Kcnk9, Cacna1c, Nrxn3, Pkhd1, Kcnd3, Esr1, Asic2, Agr2, Alb, Rbm4, Prlr, Slitrk5, Kcnip1, Nos1, Retn, 4932438A13Rik, Crhbp, Ptpn5, Heg1, Oit1, Trpc6, Htr1b, Plxna4, Sytl5, Stxbp5l, Dmxl1, Slc8a1, Ssh1, Wnk3, Tacr3, Myrip, Omg, Palm2, Tmsb10, Scgb1a1, Slc30a8, Gal, Nop10, Rims4, Cacna1e, Grid2, Kcnh5, Lnpep, Gpam, Pcsk6, Steap3, Sesn3, Gnat2, Kdelr3, Slc5a7, Rcor1, Scnn1g, Sorl1, Neb, Atp8b3, Usp13, Chrna9, Ptch1, Flrt1, Adnp, Grm5, Slc9a7, Cckar, Ppargc1b, Gckr, G6pdx, Il2ra, Ank1, Pycr1, Itga1, Cela2a, Grin2a, Brwd3, Slc24a2, Tenm1, Atp7b, Txndc2, Gng3, Vps13d, Tmod2, Cbl, Igsf9b, Ptpro, Lyst, Vkorc1, Myl4, Chst8, Hmbox1, Prl, Prkdc, Ar, Slc4a5, Tyms, Nr3c2, Gpr21, Mbd5, Stat5a, Cacna1g, Slc30a7, Tnks, Scd1, Pak3, Oprk1, Akap6, Col1a2, Cartpt, Cplx1, Syt6, Kcnj6, Slc26a2, Klhl3, Syndig1, Adgrb3, Pclo, Nme2, Malat1, Epha5, Cdkl5, F8, Fstl4, Nrip1, Uprt, Aqp7, Arhgap5, Rab38, Alms1, Atm, Ern1, Grin2b, Rab11fip2, Reln, Slc8a3, and Scn10a |
| GO:0007610~behavior | 41 | 5.13 | 4.32E-4 | Crhbp, Ptpn5, Aff2, Htr1b, Vip, Mif, Tacr3, Oprk1, Hipk2, Grin2a, Pgr, Slc24a2, Cartpt, Gal, Nhlh2, Tnr, Cacna1c, Nrxn3, Dmrta1, Tmod2, Cacna1e, Adgrb3, Strn, Uba6, Grin2b, Prl, Ar, Reln, Adcy8, Unc79, Alb, Gpr21, Adnp, Mbd5, Slitrk5, Slc8a3, Pak7, Grm5, Nos1, Retn, and Cckar |

## Analysis of estrus cycle

Vaginal smears of WT and KO mice were performed in NaCl 9/1000, fixed in methanol, and then stained with Giemsa dye. Slides were scanned with a nanozoomer (Hamamatsu).

## Whole mounts of mammary glands

Mammary glands were spread on glass slides and fixed in Carnoy's fixative for 4 h. After rehydration, the glands were stained with carmine alum stain overnight. The glands were then cleared in xylene for several weeks to eliminate the fat.

## RNA extraction and reverse transcriptase, quantitative PCR

Total RNA was isolated using TRIzol reagent (Thermo Fisher Scientific), as described by the manufacturer. Reverse transcription was performed with 1 µg of total RNA using random primers and with M-MLV enzyme (Thermo Fisher Scientific). Real-time quantitative PCR was realized with SYBR green Master Mix (Roche), on a

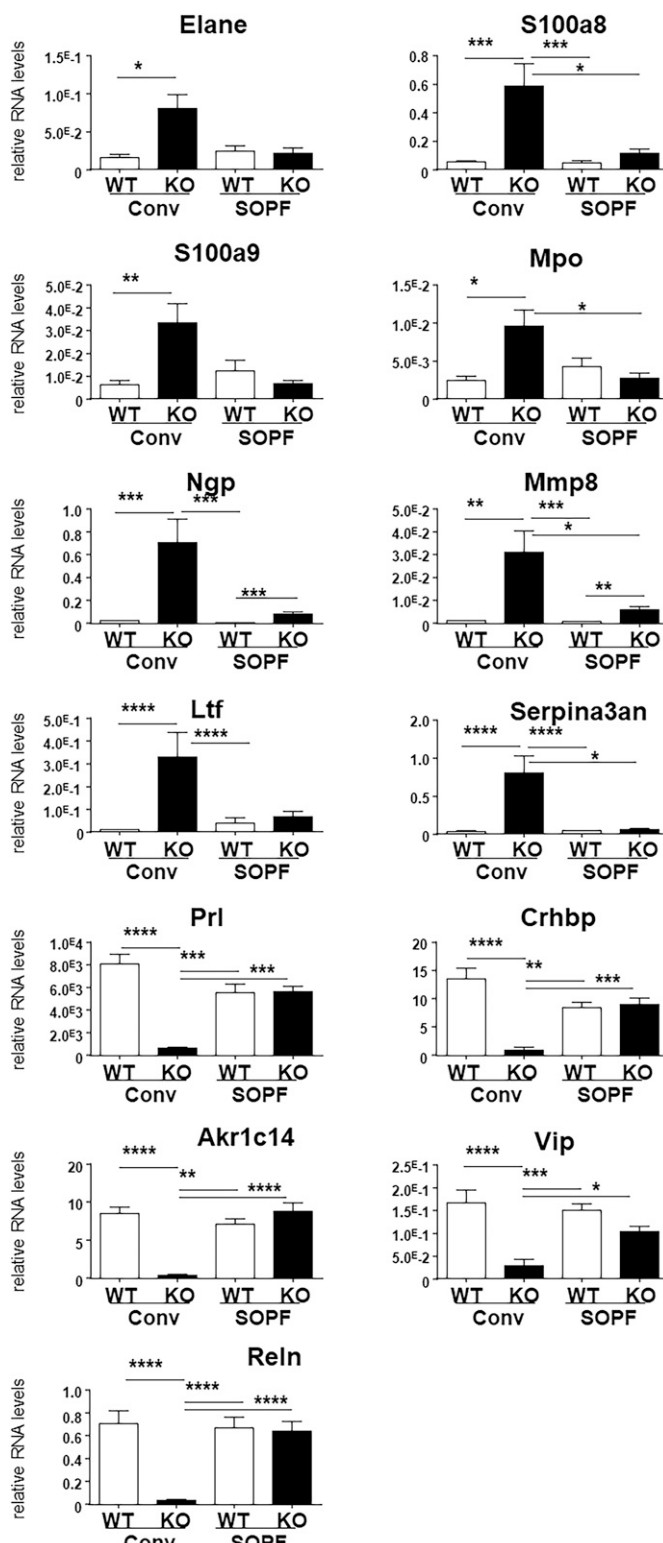

**Figure 9. Gene expression in the pituitary of KO animals in conventional housing conditions.**
Measure of RNA levels by real-time PCR of a set of genes in the pituitary of WT and *Cxcr2* KO animals in conventional or SOPF housing conditions. Results represent the mean the mean ± SEM of at least 12 animals. Kruskal–Wallis test was used followed by Dunn's multiple comparisons test to compare all groups between each other.

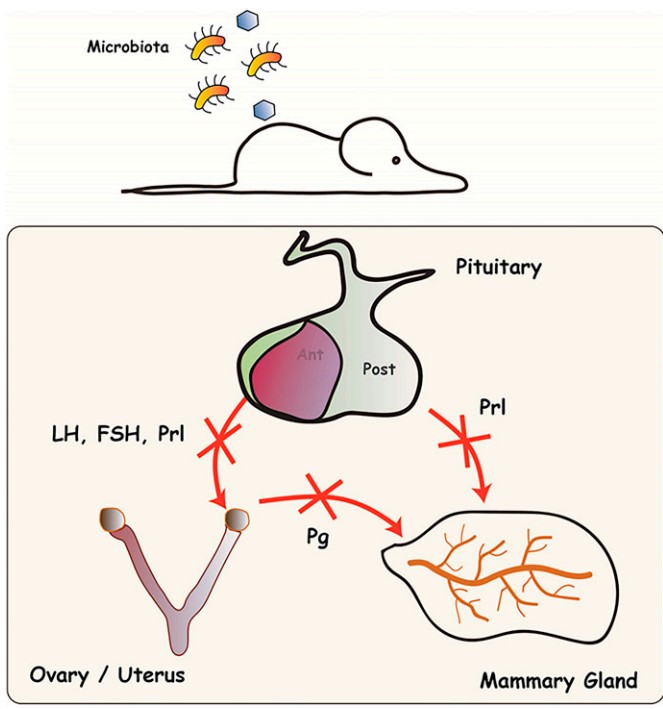

**Figure 10. General scheme of events occurring in *Cxcr2* KO mice under pathogen infection.**
Infection of *Cxcr2* KO animals by microbiota (viruses or bacteria) leads to an alteration of the pituitary function, with a defect in the production of pituitary hormones such as LH, FSH, and Prl. Low levels of these hormones impairs the function of the ovary (with an absence of ovulation) and production of progesterone. In turn, both uterus anatomy and the development of the mammary gland are affected.

Light Cycler 480 instrument (Roche) as previously described (Escobar et al, 2015). Ribosomal protein S9 (rS9) and GAPDH were used as an internal control. The sequence of the primers used in this study is indicated in Table S2. Results are expressed as N-fold differences in target gene expression relative to the internal control gene and termed "mRNA expression," were determined as mRNA expression = $2^{-\Delta Ctsample}$, where the ΔCt value of the sample was determined by subtracting the Ct value of the target gene from the Ct value of the average of the internal control genes. Target genes were considered to be non-detectable when the Ct value was above 35.

### RNAseq data processing

RNA integrity and quality was verified using RNA ScreenTape kit and Tapestation 2200 apparatus from AGILENT. cDNA libraries were synthesized using SureSelect Strand-Specific mRNA library prep (Agilent). Library quality was checked on Tapestation 2200 apparatus from AGILENT with DNA 1000 ScreenTape. Samples were sequenced on NextSeq 500 (Illumina) with an average sequencing depth of 30 millions of paired-end reads. Length of the reads was 75 bp. Each 12 Plex sample was sequenced on one HighOutput FlowCell (2 × 400 Millions of 75 bases reads). Raw sequencing data were quality-controlled with the FastQC program. Low-quality reads were trimmed or removed using Trimmomatic (minimum length: 40

bp). Reads were aligned to the mouse reference genome (mm10 build) with the TopHat2 tool (option for no multi-hits). Mapping results were quality-checked using RNA-SeQC. Gene counts were obtained by read counting software featureCounts. Normalization and differential analysis were performed with the DESeq2 package with Benjamini–Hochberg false discovery rate multiple testing correction ($P$-value < 0.05; 1.5-fold or higher change) comparing WT and KO animals. Clustering analysis was performed with Genesis software (Sturn et al, 2002). The data discussed in this publication have been deposited in NCBI's Gene Expression Omnibus (Edgar et al, 2002) and are accessible through GEO Series accession number GSE144231 (https://www.ncbi.nlm.nih.gov/geo/query/acc.cgi?acc=GSE144231).

### Hormone assessment

Whole sera were used to measure the levels of estradiol and progesterone by Elisa with specific kits from Abnova. The levels of LH, FSH, GH, and PRL were assessed by luminex with Mouse Pituitary Magnetic Bead Panel (Merck Millipore).

### Ovary and mammary transplantations

For ovary transplantations, WT or KO ovaries were obtained from 10-wk-old donor mice. WT recipients were anesthetized with isoflurane and ovariectomized. One donor ovary was implanted in the recipient's ovarian bursa. After peritoneal and skin suture, recipients were allowed to recover for 2 wk. Ovaries were then analyzed at that stage or recipient mice were mated with WT males to evaluate fertility. For mammary gland transplantation, 12-wk-old WT and KO donor mice were used to obtain mammary glands. Small pieces of donor glands were implanted in the cleared mammary fat pad of one gland of 2.7-wk-old WT mice under general anesthesia. The contralateral mammary gland was cleared and kept as a control for clearing. Recipient mice were euthanized at 12 wk and the transplanted mammary glands were analyzed by whole mount.

### Statistics

Statistical analyses were carried out using unpaired Mann–Whitney test or when indicated Fisher exact test or Kruskal–Wallis test followed by Dunn's multiple comparisons test. To assess biological interpretation of the most differentially expressed genes, GO enrichment analysis was performed using the DAVID bioinformatics resources (https://david.ncifcrf.gov, Version 6.8).

### Data Availability

The datasets discussed in this publication have been deposited in NCBI's Gene Expression Omnibus (Edgar et al, 2002) and are accessible through GEO Series accession number GSE144231 (https://www.ncbi.nlm.nih.gov/geo/query/acc.cgi?acc=GSE144231).

## Supplementary Information

## Acknowledgements

This work was supported by Association pour la Recherche sur le Cancer (ARC) Foundation and la Ligue contre le Cancer to G Lazennec and by an Agence Nationale de la Recherche (ANR) grant 17-CE14-0019 to K Balabanian. We thank N Binart, A Richmond, and M Gary-Bobo for useful discussion and help in the experiments and M/L/Loyalle and A Pulido for their help in mouse husbandry. We are grateful to J Stingl for training in mammary gland transplantation. We acknowledge the Plateau Central d'Elevage et d'Archivage (PCEA), Réseau des animaleries de Montpellier (RAM), Montpellier Ressources Imagerie (MRI), and Setiph facilities in Montpellier. We are grateful to Institut du cerveau et de la moëlle épinière in Paris for RNAseq experiments. We thank CECEMA-UM animal facility for housing and technical help. We are grateful to all master students who have contributed to this work.

### Authors Contributions

C Timaxian: investigation.
I Raymond-Letron: investigation and methodology.
C Bouclier: investigation.
L Gulliver: investigation—review and editing.
L Le Corre: investigation.
K Chebli: investigation and methodology.
A Guillou: investigation.
P Mollard: writing—review and editing.
K Balabanian: writing—review and editing.
G Lazennec: conceptualization, supervision, investigation, methodology, and writing—original draft, review, and editing.

### Conflict of Interest Statement

The authors declare that they have no conflict of interest.

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
