## [Reviewer comments · Life Science Alliance]

The health status alters the pituitary function and reproduction of mice in a *Cxcr2*- dependent manner

Colin Timaxian, Isabelle Raymond-Letron, Celine Bouclier, Linda Gulliver, Ludovic Le Corre, Karim Chebli, Anne Guillou, Patrice Mollard, Karl Balabanian, and Gwendal Lazennec

DOI: 10.26508/lsa.201900599

Corresponding author(s): Gwendal Lazennec, CNRS

Review timeline:

Submission Date:	2019-11-07
Editorial Decision:	2019-12-04
Revision Received:	2020-01-14
Editorial Decision:	2020-01-21
Revision Received:	2020-01-28
Accepted:	2020-01-29

Scientific Editor: Andrea Leibfried

Transaction Report:

1st Editorial Decision

04 December 2019

December 4, 2019

Re: Life Science Alliance manuscript #LSA-2019-00599-T

Dr. Gwendal Lazennec
CNRS
UMR9005 Sys2Diag
1682 rue de la Valsiere
Montpellier 34184
France

Dear Dr. Lazennec,

Thank you for submitting your manuscript entitled "The microbiome alters the pituitary function and reproduction capacities of mice in a CXCR2-dependent manner" to Life Science Alliance. The manuscript was assessed by expert reviewers, whose comments are appended to this letter.

As you will see, the reviewers appreciate your manuscript and its value to the community, and they provide constructive input on how to further strengthen your work. We would thus like to invite you to submit a revised version of your manuscript to us, addressing the criticisms raised. Please get in touch in case you would like to discuss individual revision points raised by reviewer #2 further with us prior to embarking into the experimental revision.

Thank you for this interesting contribution to Life Science Alliance. We are looking forward to receiving your revised manuscript.

Sincerely,

Andrea Leibfried, PhD
Executive Editor
Life Science Alliance
Meyerhofstr. 1
69117 Heidelberg, Germany

t +49 6221 8891 502
e a.leibfried@life-science-alliance.org
www.life-science-alliance.org

B. MANUSCRIPT ORGANIZATION AND FORMATTING:

Reviewer #1 (Comments to the Authors (Required)):

1. This manuscript makes the novel discovery that the microbiota differentially affects expression of genes in mice with global loss of CXCR2 as compared to those without loss of CXCR2. Of specific interest are alterations in expression of pituitary hormones that regulate the reproductive tract and mammary gland function of female mice. While it has been known for some time by those investigators working with CXCR2 null mice that it is challenging to breed CXCR2 null mice in conventional animal housing conditions, no one has ever done the work of characterizing the reasons for this defect. This paper clearly shows that the defect is linked to the combined effect of the microbiota and loss of CXCR2 and when the CXCR2 null ovary or mammary tissue is transplanted to WT mice, under standard housing conditions, the development of the ovary and mammary gland is normal. Thus it is the effect of loss of CXCR2 on the microbiome that creates this effect leading to changes in pituitary secretion of FSH, LH, PRL and GH that bestows the phenotype of the CXCR2 null mice when housed

under standard housing conditions. In SOPF conditions, the phenotype of the CXCR2-WT and CXCR2-null mice are equivocal.

2. This paper also investigates the global changes in gene expression in the CXCR2-WT vs CXCR2-null mice under SOPF versus standard housing conditions. They show that in the absence of CXCR2, bystander infections affect leukocyte migration, adhesion and function, as well as ion transport and synaptic function behavior. The data are well documented with PCR analysis in follow-up of the GO-analysis of the transcriptomes of these mice.

3. Of particular interest CXCR2-null mice in standard conditions exhibit down-regulation of lactoferrin, GATA3, cyclin D1, Erbb2, Elf5, Epcam, Prlr, Krt15, Epcam, Wnt4, and Wnt5b. Of note Prlr, cyclin D1, Erbb2 and Elf5 are crucial for mammary gland development as shown by other groups. Moreover Wnt signaling activates ERa-positive luminal cells--that then are stimulated by ER to release Wnt ligands that act on the myoepithelial compartment.

4. The pituitary of CXCR2 KO mice exhibited T lymphocyte enrichment and to a lesser extend B lymphocyte, macrophage and neutrophil enrichment. This type of inflammatory environment is similar to that of autoimmune hypophysitis that can led to hypo-pituitarism and atrophy of the pituitary. This finding is discussed thoroughly in the discussion and adds to the significance of the study. Importantly it has implications for the use of CXCR2 antagonist side effects with long term use--though this point is not made.

5. Another important point is the relevance of studies of mice in SOPF conditions. Based upon the findings reported here, it is clear that there will be many differences in gene expression that limit the use of studies done in SOPF conditions--since humans are never in SOPF conditions.

6. Some edits may be needed in the wording of the manuscript--but this is a minor concern.

Reviewer #2 (Comments to the Authors (Required)):

The study by Timaxian et al. characterizes the pituitary function and reproductive abilities of SOPF and conventionally raised (CONV) female mice, especially in the context of a genetic invalidation of *Cxcr2* chemokine.

The results of the study are the following:

- a. Under CONV, but not SOPF conditions, *Cxcr2*-null female mice are unable to cycle normally, affecting their reproductive capability
- b. Under CONV, but not SOPF conditions, the function and histology of uterus and mammary gland of *Cxcr2*-null females show abnormalities
- c. Transplanting the ovary or mammary gland of KO mice into WT mice under CONV conditions recapitulates the phenotype observed in KO mice
- d. Pituitary-gonadal axis signaling is altered in *Cxcr2*-null females under CONV, but not SOPF, conditions.

Overall, the conclusions of the study are well supported. These are very novel findings, which come in a timely manner where more and more studies aim to understand the impact of health status on the physiology of laboratory mice. I especially appreciated the transplantation experiments, which show a clear phenotype linked to the ovary of mammary gland of KO mice.

I do have a few comments, however.

MAJOR:

1. The title of the manuscript should be changed, along with most references to "microbiota". While we can infer that the observed phenotypes are the consequence of the microbial environment, this does not necessarily mean that it is by the microbiota. I would replace "microbiome" in the title by "health status". If the authors want to claim that the microbiota is the culprit, I would suggest two things:
 - a. Switch the bedding of SOPF and CONV mice (and vice-versa). As I understand that CONV bedding cannot be brought into the SOPF room, I'd suggest that SOPF mice be brought to the CONV room and bedding be switched between CONV and SOPF mice.
 - b. Performing fecal microbiota transplantation from CONV to SOPF and vice-versa. Then the role of the role of the microbiota will be clear.
2. Since Cxcr2-null mice are used, I think that it would be very useful to have the levels of the chemokine in WT CONV and SOPF mice. What is the rationale to focus on that chemokine specifically?
3. What is the rationale to focus solely on female mice? I understand that adding male mice to the analysis doubles the amount of work, but there should be an explanation to it. Did the authors perform semen analysis of male mice?
4. Why is SOPF data absent in several figures (such as Figure 2A)? That would be extremely important.
5. Authors should quantify the data when that is possible (e.g. branching in mammary gland in Figure 6D and luteal bodies in Figure 6C). In the same manner, I'd suggest merging the quantification data from supplementary Figure 2 into Figure 3. If qualitative data (images) can be quantified, it should be done.
6. In page 6, the authors claim that "rederivating conventional animals to remove pathogens led to animals with full reproduction ability". The data needs to be shown.
7. In Figures 4 and 7, a multilinear model would be very useful. Adding a PCoA plot would add to the understanding of the study, if the plot shows clear segregation in CONV but not SOPF conditions. In Fig. 7E, two-way ANOVA should be used to include the two variables: health status and genotype (with adjusted p-values). Chi-2 or Fisher exact test should be done in Figure 6C.
8. In transplanted animals, at least hormone levels should be measured.
9. Can the authors imagine that stimulating KO mice in CONV conditions with LH or FSH would restore the WT phenotype? If this cannot be done, at least it should be discussed.

MINOR COMMENTS:

1. Nomenclature of genes should be respected for the species. In mice, the nomenclature of a gene is Cxcr2, with only the first letter being capital.
2. The authors should include in the discussion the paper by Rosshart et al. (Science, 2019), showing that laboratory mice born to wild mothers retain certain immunological abilities.
3. In order for the figures to appear easier to read, I'd suggest adding CONV or SOPF as headers whenever those mice are used. GO term figures are very difficult to read and understand. Can the authors provide a table instead?

4. I suggest merging Figure 6E into Figure 7.

Reviewer #1 (Comments to the Authors (Required)):

Answer: There is no specific point raised by the reviewer except in the wording, that we have done our best to improve.

Reviewer #2 (Comments to the Authors (Required)):

1. The title of the manuscript should be changed, along with most references to "microbiota". While we can infer that the observed phenotypes are the consequence of the microbial environment, this does not necessarily mean that it is by the microbiota. I would replace "microbiome" in the title by "health status". If the authors want to claim that the microbiota is the culprit, I would suggest two things:

- a. Switch the bedding of SOPF and CONV mice (and vice-versa). As I understand that CONV bedding cannot be brought into the SOPF room, I'd suggest that SOPF mice be brought to the CONV room and bedding be switched between CONV and SOPF mice.
- b. Performing fecal microbiota transplantation from CONV to SOPF and vice-versa. Then the role of the role of the microbiota will be clear.

Answer: We agree with the reviewer comment that it could be a bit confusing. Indeed, SOPF animals were brought to a conventional room and to favor the rapid acquisition of microbiota from the facility, the bedding of animals already in conventional conditions was added to the one of transferred SOPF animals during the first weeks. We have added this to the materials and methods. We have not done fecal transplantations, but the added bedding is close to that. The reviewer is also right, conventional animals cannot go to a SOPF facility. So, we have replaced in the title microbiome by health status and in the manuscript the term microbiome by microbiota.

2. Since *Cxcr2*-null mice are used, I think that it would be very useful to have the levels of the chemokine in WT CONV and SOPF mice. What is the rationale to focus on that chemokine specifically?

Answer: Our work starts with the serendipitous discovery that we observed specifically on *CXCR2* KO animals the loss of fertility when SOPF animals were transferred to a conventional facility. To our knowledge, no one has reported this for any other chemokine or chemokine receptor KO animal. The question of the reviewer about chemokines is interesting but is also relatively broad. We looked at the levels of *CXCR2* ligands in the mammary gland and pituitary of WT animals in conventional and SOPF conditions based on RNAseq analysis. From this analysis, we did not find any significantly regulated *CXCR2* ligands between WT animals in the mammary gland of conventional and SOPF conditions. The same was true for WT pituitaries of conventional and SOPF animals. We have mentioned it in the discussion of the manuscript. Regarding other chemokines, only the chemokines *CCL6* and *CCL8* are down-regulated in the mammary gland of WT SOPF animals compared to CONV animals.

3. What is the rationale to focus solely on female mice? I understand that adding male mice to the analysis doubles the amount of work, but there should be an explanation to it. Did the authors perform semen analysis of male mice?

Answer: The reviewer is right, looking at males animals would require an entire study by itself and may dilute the message of this paper. What we can say is that CXCR2 KO males are also infertile. They also show a reduced size of prostate and seminal glands. We plan to study this more in details in the future.

4. Why is SOPF data absent in several figures (such as Figure 2A)? That would be extremely important.

Answer: SOPF data are present in most of the figures and as the differences between WT and animals were only observed in conventional conditions, we did not include this in figure 2. We did not see any alteration of reproduction nor of ovary or uterus histology of SOPF animals as shown in figure 2B and 3A. We have now added SOPF smears to the manuscript. To limit the size of the figure 2, we have added the SOPF smears as a supplementary figure 2.

5. Authors should quantify the data when that is possible (e.g. branching in mammary gland in Figure 6D and luteal bodies in Figure 6C). In the same manner, I'd suggest merging the quantification data from supplementary Figure 2 into Figure 3. If qualitative data (images) can be quantified, it should be done.

Answer: As suggested by the reviewer, we have included quantification of corpora lutea in the ovaries for fig 6C. But to our knowledge, there is no way to quantify the branching on whole mounts pictures of the mammary gland, this has not been done in previous papers studying mammary gland function (see for instance Brisken et al, 2000, Dev Cell from R.A.Weinberg's group or Feng et al., 2007, PNAS from S.A. Khan lab).

We have merged supplementary figures 2 and 3 but the idea of separating them was that Supp Fig 2 corresponds to uterus and Supp Fig 3 to mammary gland, to avoid any possible confusion. This merged figure is now Supplementary figure 3.

6. In page 6, the authors claim that "rederivating conventional animals to remove pathogens led to animals with full reproduction ability". The data needs to be shown.

Answer: We have effectively rederivated conventional animals to transfer them into a SOPF facility. We have never been able to have pups from homozygous animals in conventional facility. On the contrary, all homozygous CXCR2 KO animals which were rederived from conventional animals that we have mated in SOPF conditions were fertile. This is what is shown in Fig. 1A, these are animals in SOPF conditions after having been rederivated from conventional animals.

7. In Figures 4 and 7, a multilinear model would be very useful. Adding a PCoA plot would add to the understanding of the study, if the plot shows clear segregation in CONV but not SOPF conditions. In Fig. 7E, two-way ANOVA should be used to

include the two variables: health status and genotype (with adjusted p-values). Chi-2 or Fisher exact test should be done in Figure 6C.

Answer: We have added a supplementary figure 4 that shows the PCA for both mammary gland and pituitary RNAseq. This shows that the transcriptome of WT and KO animals are much more different in conventional conditions than in SOPF conditions both in the mammary gland and the pituitary. Pituitary and mammary gland samples are clearly different as expected. Finally, without taking in account the genotype of the animals (WT or KO), the transcriptomes of conventional animals are also different from the one of SOPF animals both in the pituitary and the mammary gland.

In figure 7E (now labelled 7F), we have re-run statistical analysis and we have used Kruskal-Wallis test followed by Dunn's multiple comparisons test to compare all plots between each other. This confirms that the main differences are between WT and KO animals in CONV conditions. In addition, statistical differences are seen for some genes between KO from CONV and SOPF conditions, but not between WT animals.

We have performed a Fisher exact test on Fig. 6C, and added it to the figure legend, this shows no significant difference in successful breeding between WT and KO donors ($p=0.5147$).

8. In transplanted animals, at least hormone levels should be measured.

Answer: The main goal of doing ovarian transplantations was to show that the WT females transplanted with KO ovary were fertile and to show that the histology of transplanted KO ovaries was normal in WT environment. For these reasons, transplanted females were mated after transplantation and when they gave birth, the females were sacrificed to check the ovary histology. We did not collect blood samples for these animals as they had been pregnant (and had thus obviously a very particular status) and could not be compared with any of the animals of this study assessed before which were all virgins.

9. Can the authors imagine that stimulating KO mice in CONV conditions with LH or FSH would restore the WT phenotype? If this cannot be done, at least it should be discussed.

Answer: Treating mice with LH or FSH is a good suggestion but these hormones are acting in a cyclic manner. So to be effective, this would require to have some kind of pulsatile injections at correctly chosen times according to the estrus cycle. The problem is that these animals have a weird estrus cycle, so, there is no way to know when these animals should be treated synchronously. Moreover, progesterone and estradiol levels are altered and the histology of the uterus and ovary severely impaired, it might not work. We have now discussed this in the paper.

MINOR COMMENTS:

1. Nomenclature of genes should be respected for the species. In mice, the nomenclature of a gene is *Cxcr2*, with only the first letter being capital.

Answer: Thank you for the remark, we have modified this in the manuscript

2. The authors should include in the discussion the paper by Rosshart et al. (Science, 2019), showing that laboratory mice born to wild mothers retain certain immunological abilities.

Answer: We have included this reference in the introduction of the paper, this is where we introduced the notion of microbiota effects on mouse phenotypes, the discussion being more focused on the understanding of *Cxcr2* phenotype.

3. In order for the figures to appear easier to read, I'd suggest adding CONV or SOPF as headers whenever those mice are used. GO term figures are very difficult to read and understand. Can the authors provide a table instead?

Answer: We have added CONV and SOPF in all figures, whenever possible and not surcharging the figure.

Concerning the GO terms, the list of the most enriched pathways is given in tables 1 to 6. The interest of showing a figure is to show the connections between the related GO terms, which cannot be shown in the tables. Moreover, these figures enable to show connections with pathways that are less enriched and have not been put in the tables to avoid a huge list of GO terms. So, we believe that having both the figures of the GO and the GO tables of the most enriched GO gives the best view of the GO analysis.

4. I suggest merging Figure 6E into Figure 7.

Answer: We have merged 6E into 7 (it is now 7A) as suggested by the reviewer

2nd Editorial Decision

21 January 2020

January 21, 2020

RE: Life Science Alliance Manuscript #LSA-2019-00599-TR

Dr. Gwendal Lazennec
CNRS
UMR9005 Sys2Diag
1682 rue de la Valsiere
Montpellier 34184
France

Dear Dr. Lazennec,

Thank you for submitting your revised manuscript entitled "The health status alters the pituitary function and reproduction of mice in a Cxcr2-dependent manner". As you will see, reviewer #2 largely appreciates the introduced changes. We would thus be happy to publish your paper in Life Science Alliance pending final minor revisions:

- Please address the reviewer's remaining concerns
- Please deposit the RNA-seq data in a repository and provide the accession number in the methods section of your manuscript
- Please add the statistical test used in the figure legends wherever mentioning p-values
- Please note that figures are displayed in the published version of your manuscript the way you provide them - Figures 4, 5 and 7 are too difficult to read - please provide versions with increased font size to allow readers to appreciate the content at the actual size of the figure
- Please add callouts in the manuscript text to Fig 8 and S1
- Please list in your reference list 10 authors et al
- Please add scale bars to your figure panels (those already shown cannot be properly read)

A. FINAL FILES:

-- High-resolution figure, supplementary figure and video files uploaded as individual

files: See our detailed guidelines for preparing your production-ready images,
<http://www.life-science-alliance.org/authors>

B. MANUSCRIPT ORGANIZATION AND FORMATTING:

Sincerely,

Andrea Leibfried, PhD
Executive Editor
Life Science Alliance
Meyershofstr. 1
69117 Heidelberg, Germany
t +49 6221 8891 502
e a.leibfried@life-science-alliance.org
www.life-science-alliance.org

Reviewer #2 (Comments to the Authors (Required)):

In this revised version of the manuscript, Timaxian et al. have incorporated several modifications and taken into account my previous comments. Overall, I feel that the manuscript has greatly improved. However, I still have a few suggestions which, to my sense, could be taken into consideration by modifying or adding a few statements in the manuscript.

1. My main concern when reading the manuscript is the rationale behind Cxcr2. I understand from the authors' response that this was a serendipitous finding, as is often the case in science. I would suggest discussing about the genesis of this random finding. Why did you study Cxcr2 to begin with? Beside the scientific background, I always appreciate reading the backstory of this scientific discovery.
2. In Figures 4 and 7, the authors have added volcano plots, not PCA. I think PCA would be useful to visualize how, as a whole, SOPF and CONV mice cluster, and how much variance is present in each axis. This should not be too complicated to do.
3. Since hormones were not measured in transplanted animals, and I understand this is not feasible any longer, I would suggest adding a sentence about this.

Dear Dr Leibfried,

Please find attached a revised version of our revised manuscript #LSA-2019-00599-T entitled "The health status alters the pituitary function and reproduction of mice in a Cxcr2-dependent manner", which we submit to Life Science Alliance for consideration for publication.

Here are the novel modifications:

- We provide also a rebuttal letter with the answers to the reviewers.
- We have deposit the RNA-seq data in GEO and have provided the accession number in the methods section of your manuscript (GSE144231).
- We have added the statistical tests in the figure legends.
- We have increased font sizes and the figures themselves whenever possible to a better reading. We have split some figures to allow a better resolution.
- We have added callouts in the manuscript text to Fig 8 (now named Fig 10) and S1 supplemental figure.
- We have listed in our references list 10 authors et al.
- We have made the scale bars bigger in the figures to allow a better visualization.

January 29, 2020

RE: Life Science Alliance Manuscript #LSA-2019-00599-TRR

Dr. Gwendal Lazennec
CNRS
UMR9005 Sys2Diag
1682 rue de la Valsiere
Montpellier 34184
France

Dear Dr. Lazennec,

Thank you for submitting your Research Article entitled "The health status alters the pituitary function and reproduction of mice in a Cxcr2-dependent manner". It is a pleasure to let you know that your manuscript is now accepted for publication in Life Science Alliance. Congratulations on this interesting work.

DISTRIBUTION OF MATERIALS:

Again, congratulations on a very nice paper. I hope you found the review process to be constructive and are pleased with how the manuscript was handled editorially. We look forward to future exciting submissions from your lab.

Sincerely,

Andrea Leibfried, PhD
Executive Editor

Life Science Alliance
Meyerohofstr. 1
69117 Heidelberg, Germany
t +49 6221 8891 502
e a.leibfried@life-science-alliance.org
www.life-science-alliance.org